# Structural Batteries: A Review

**DOI:** 10.3390/molecules26082203

**Published:** 2021-04-11

**Authors:** Federico Danzi, Rui Martim Salgado, Joana Espain Oliveira, Albertino Arteiro, Pedro Ponces Camanho, Maria Helena Braga

**Affiliations:** 1LAETA, Department of Engineering Physics, Engineering Faculty, University of Porto, Rua Dr. Roberto Frias, s/n, 4200-465 Porto, Portugal; jespain@fe.up.pt; 2INEGI, Instituto de Ciência e Inovação em Engenharia Mecânica e Engenharia Industrial, Rua Dr. Roberto Frias, 400, 4200-465 Porto, Portugal; aarteiro@fe.up.pt; 3DEMec, Faculdade de Engenharia, Universidade do Porto, Rua Dr. Roberto Frias, 4200-465 Porto, Portugal; up201603995@fe.up.pt

**Keywords:** structural batteries, solid electrolyte, composite materials and multifunctional materials

## Abstract

Structural power composites stand out as a possible solution to the demands of the modern transportation system of more efficient and eco-friendly vehicles. Recent studies demonstrated the possibility to realize these components endowing high-performance composites with electrochemical properties. The aim of this paper is to present a systematic review of the recent developments on this more and more sensitive topic. Two main technologies will be covered here: (1) the integration of commercially available lithium-ion batteries in composite structures, and (2) the fabrication of carbon fiber-based multifunctional materials. The latter will be deeply analyzed, describing how the fibers and the polymeric matrices can be synergistically combined with ionic salts and cathodic materials to manufacture monolithic structural batteries. The main challenges faced by these emerging research fields are also addressed. Among them, the maximum allowable curing cycle for the embedded configuration and the realization that highly conductive structural electrolytes for the monolithic solution are noteworthy. This work also shows an overview of the multiphysics material models developed for these studies and provides a clue for a possible alternative configuration based on solid-state electrolytes.

## 1. Introduction

Lithium-ion based batteries have already demonstrated an unparalleled combination of high energy and power density, quick charge and long-life that made this technology the present choice for electric vehicles, portable electronic devices, and many other applications. In recent years, the research in this field moved companies and governments to invest an extensive amount of money for developing more and more efficient and safer types of batteries. Yet looking at the future, advanced and green energy storage components are nowadays the main drivers for all modern transportation systems. From electric vehicles (EVs) to unmanned aerial vehicles (UAVs), from high altitude satellites to electric airplanes, the target for the next years is the development of lighter, greener, and more sustainable vehicles. A great opportunity to achieve all of these goals at the same time is offered by the use of multifunctional systems and materials [1,2,3,4]. As described by Thomas et al. [5], multifunctional systems and materials correspond to all the structural configurations and materials that are capable of fulfilling more than one primary function simultaneously. In this perspective, the idea of manufacturing structural composite batteries capable of storing electric energy and, at the same time, carrying mechanical loads is one of the most appealing applications of multifunctionality.

Two main approaches have already been investigated. The first one focuses on adding functionalities to structures by embedding off-the-shelf thin batteries into composite laminates or sandwich panels [5,6,7,8,9,10,11,12]. The other one aims to realize multifunctional composite materials where the reinforcement elements act as the electrodes while the polymeric matrix works as the electrolyte and as a structural binder for the fibers [13,14,15,16,17,18,19]. An alternative path could come from the use of all-solid-state electrolytes. Figure 1 shows the basic conceptualization of these alternatives, while a schematic representation of the physical principles and the constructive solutions described in this paper is shown in Figure 2.

The embedded cell idea emerged from the necessity of optimizing the volume, more than the weight, of a composite structure by embedding electrical power elements without compromising their mechanical performance. The bonding process, anyway, does not result in a remarkable overall improvement due to the fact that the battery elements, as they are, are bearing no load, hence their mass is not contributing at all to the structural performance of the final product. The other concept of monolithic multifunctional materials, instead, comes from the consideration that high-performance composites and modern lithium-ion batteries have several features in common. First, the fact that carbon fibers, commonly used in high-performance composites for their high specific stiffness and strength, also exhibit remarkable electrochemical properties such as good electrical conductivity and high lithium-ion intercalation in the case of the graphite carbon allotrope. In a second development, the layered configuration that characterizes both modern composites and state-of-the-art lithium-ion batteries could be exploited for a synergistic design. Moreover, the well-known wide range of composites processing techniques enables great freedom in the design of innovative configurations suitable also for the structural batteries.

Unfortunately, as in all the other technological progress, the development and the manufacturing of this new class of materials poses new challenges for the researchers. Concerning the embedding alternative, the main drawback comes from the maximum operating temperature of the power elements, which is usually <60 °C [22]. This limit is far below the typical curing temperatures of high-performance composites and adhesives, commonly above 100 °C. For this reason, a co-curing of the batteries in high-performance laminates in compliance with the material curing cycle is unfeasible. Available alternatives, such as the bonding of the pre-cured laminates with the batteries or co-curing using low temperature curing resin systems, still remain a valid option, and these recent ideas are described in this work.

Regarding the multifunctional materials, however, the main challenges come from the swelling and the shrinkage of the electrodes during the lithiation and delithiation process, the synthesis of electrochemically efficient solid electrolytes, and the development of structural positive electrodes. The first challenge is critical for the evaluation of the stress field produced during the battery operation, which has to be taken into consideration during the design phase to avoid the formation of micro damages. The realization of a solid electrolyte, however, is of central importance to guarantee safety and a separator/electrolyte integrated solution without undermining the mechanical performance of the composites, while the development of structural cathodes is an essential technological step to realize efficient energy storage composites.

Whether it is the integration of state-of-the-art available batteries in composite structures or the formulation of new monolithic structural materials, a great step forward still has to be done to bring structural energy storage devices to the market.

In this paper, the review of multifunctional systems and materials used for manufacturing structural energy composites is presented. The main concepts for the development of structural energy storage devices, corresponding to the multifunctional systems and multifunctional materials shown in Table 1, will be discussed. A preliminary overview of the theoretical framework created to evaluate the performance of multifunctional configurations is followed by a description of the main progress in embedding batteries in composite structures. The following sections are dedicated to the recent findings in the development of monolithic multifunctional composite materials and the application of solid-state electrolytes in structural power composites. A brief introduction is followed by a description of the main progress in the development of new alternative electrodes and electrolytes. Finally, the recent findings in terms of analytical and numerical methods produced for studying electric power composites are presented, as well as possible avenues for the development of structural energy storage devices.

## 2. Multifunctionality Evaluation Methods

Before discussing the details on the state-of-the-art solutions for multifunctional composite materials and systems, it is worth understanding how to evaluate the potential gain derived by embracing this innovation. The margin in optimizing a structure using materials and design solutions capable of accomplishing more than one primary function was theorized for the first time for structural power composites by Thomas et al. [30]. In their work, the idea of improving a critical vehicle parameter such as the flight endurance of an UAV was rethought considering the integration of the batteries in the structure. Starting from Ashby’s notion of ranking the performance of a material according to system-level objectives [31], the concept of material-architecture indices has been introduced and applied in the optimization of a structural battery. The idea was then realized by the same research group [23] who manufactured two multifunctional micro-UAV configurations with an embedded battery pack. These new structures, compared with the traditional configuration, proved the possibility of increasing the endurance of a vehicle by reducing the structural weight and integrating the batteries in the structure itself, as shown in Figure 3.

In the following years, O’Brien and coworkers [32] specialized the notion of multifunctional efficiency for minimizing the system mass of a device with both energy storage and structural capabilities. They introduced the idea of multifunctional efficiency, defined by Equation (1), and applied it directly to multifunctional capacitors.
(1)$$\eta_{mf}\;=\;\eta_{s}\;+\;\eta_{e}\;>\;1$$

Here it can be noticed that the multifunctional efficiency is just the algebraic sum of the structural efficiency *ηs* expressed in terms of specific stiffness of the multifunctional material (*E*¯*_m f_*) and the energy efficiency *ηe* as a function of the energy density (*w*_¯ *m f*_) as shown in Equation (2).
(2)$$\eta_{mf}\;=\;\eta_{s}\;+\;\eta_{e}$$

The new concept was then applied to evaluate the performance of composite capacitors, such as that shown in Figure 4, that were manufactured with E-glass fabrics dielectrics of different volume fractions. The experimental campaign, including the dielectric breakdown tests and tensile tests, was performed and the results used to detect the multifunctional efficiency for this class of capacitors, showing the possibility to have a consistent mass saving. More recently, Snyder et al. [33] extended the concept of multifunctional efficiency to a more general problem. In their approach, multiple structural and electrical performance was taken into account simultaneously. In particular, not only the specific elastic modulus *E*¯*_mf_*, but also the specific shear modulus *G*¯*_mf_*was considered for the estimation of the mechanical performance. On the electrical side, instead, the specific multifunctional energy *w*¯ *_mf_*and the specific power *p*¯*_mf_*were evaluated simultaneously. All these parameters were then compared with the non-multifunctional material properties to define both the structural and electrical efficiencies for the multifunctional using Equation (3).
(3)$$\eta_{s}=min\left\{\frac{\bar{E_{mf}}}{\bar{E}},\frac{\bar{G_{mf}}}{\bar{G}}\right\};\;\;\eta_{e}=min\left\{\frac{\bar{w_{mf}}}{\bar{w}},\frac{\bar{p_{mf}}}{\bar{p}}\right\}$$

Under this new light, Equation (1) can be used to evaluate the global multifunctional efficiency of the system. The conclusion of this interpretation of the multifunctional efficiency is that a mass reduction cannot be achieved just by integrating an electrical device into a structure while this is still a valid option for optimizing the volume of a structure. The idea was then practically applied for the evaluation of the multifunctional performance of composite supercapacitors. The devices were manufactured via vacuum-assisted resin transfer molding using T300 plain-weave fabric as reinforcements, while the structural electrolyte was composed of lithium salt dissolved in vinyl ester resin. From the electro-mechanical tests, the highest values of multifunctional efficiency were below 0.2, and from a retrospective analysis of several multifunctional solutions presented up to that time, a maximum value of 0.5 was estimated.

An alternative interpretation of the optimization of the multifunctional materials was proposed by Johannisson et al. [28]. This approach instead focused on calculating the mass of the structural battery *m_SB_* and compared it to the combined mass of an equivalent carbon fiber composite plate *m_CC_* and of a standard lithium-ion battery *m_LiB_*. In detail, the mass of a composite plate with equivalent stiffness to specific loading conditions and the mass of a standard battery with a certain amount of electrical energy were compared with the mass needed to manufacture a structural battery with the same performance. This concept was then used to estimate the multifunctional capabilities of relevant structures, such as an interior panel of an aircraft, an EV roof, and an electric ferry, showing potential mass savings. In a follow-up study, this concept was used by Zackrisson et al. [34] as the base for a preliminary estimation of the environmental implications related to the introduction of this technology in the EVs market. This work considered the replacement of a steel EV roof with a structural battery and analyzed all the implications on its full life cycle, including production, use, and recycling. Life cycle assessment techniques were eventually used to prove that the introduction of this technology could substantially reduce climate impact, ozone, and abiotic depletion. More recently, Johannisson et al. [35] introduced a new idea for evaluating the performance of multifunctional systems founded on the concept of residual specific properties. These parameters were estimated as the ratio of the specific physical properties required by the multifunctional system/device, divided by its residual mass. The latter was evaluated as the difference in mass between the multifunctional system and the associated monofunctional solutions. This new specific metric provides a practical tool for the scientist working in the field to weigh if the implementation of a multifunctional configuration has a real advantage compared with the monofunctional available solutions. The concept explained in this work for the mass can be easily extended to other relevant aspects, such as the cost and the volume of the system to be developed, without loss of generality.

Undoubtedly, the mass and the volume savings are the most attractive features of optimising in transportation systems, but furthermore, the battery safety, capacity, cycle life, and resilience up to a broader range of temperatures should be taken into account.

## 3. Multifunctional Systems

The recent developments in integrating and packing energy storage devices in composite structures are discussed in this section. Along the lines of the preliminary work presented on the multifunctional layout for UAVs introduced by Thomas et al. [23], the idea was used for engineering other space and marine structures with the aim of saving both mass and volume. Roberts et al. [7,36] presented multifunctional sandwich configurations with electrical storage capability for satellites. The multifunctional sandwich structures with aluminum honeycomb and carbon fiber face sheets were customized to fit market-available polymer lithium-ion cells. The cells here embedded were the Varta PoliFlex with a specific energy of about 200 W h/kg and a maximum operating temperature of 60 °C, manufactured with a graphite anode, a lithium cobalt oxide (LiCoO_2_) cathode, and a polymeric separator with (LiPF_6_) carbonate-based electrolyte. The structures were tested under thermal, vacuum, and severe vibration conditions while both the electrical and mechanical properties were qualitatively monitored. Notwithstanding the success of the idea, new technological challenges immediately popped-up. These include the limitation in the maximum allowable temperature for curing the face sheets due to the maximum allowable temperature of the battery, the need for the reduction in cells thickness for optimizing the panels, and the optimization of the battery number and distribution in the panel. A parallel work was presented by Pereira et al. [6,37] who investigated the effect of embedding commercial all-solid-state thin-film Li-ion energy cells in carbon fiber reinforced laminates. Starting from flexural and uniaxial compression tests of the cell to verify their capability of withstanding the autoclave process, Pereira and coworkers designed an innovative power composite structure. The cells, realized with a 2 μm lithium metal anode, a 50 μm lithium phosphorus oxynitride (LiPON) and a 6 μm lithium cobalt oxide (LiCoO_2_) cathode, and a 2 μm platinum collector, as shown in Figure 5a, were enclosed in a 50 μm surlyn sealant sandwiched in two 50 μm muscovite substrate for a total thickness of 150 μm. The devices, provided by Front Edge Technologies, with a nominal voltage of 3.6 V and a specific energy of 200 W h/kg, were co-cured in the autoclave with [0/90]_S_ T700SC and AS4/3502 skins at more than 120 °C and 5 bar. A schematic representation of the constructive solution used here is shown in Figure 5b. In this configuration, the structural cell had a charge capacity of about 2 mA h/g and a specific energy of about 7.2 W h/kg, which equaled 3.6% of the battery without the composite laminate. The tensile tests were performed on the manufactured laminates to evaluate the mechanical degradation caused by the embedding, while the evaluation of the energy cell capacity was performed after the curing cycle and during the mechanical test. Mechanical baseline values were obtained from the tensile test on laminates without embedding, while the reference electrical performance was measured with the charge/discharge test on the cell as-received. No deviation was recorded on the cell properties after the embedding, and, at the same time, the tensile behavior of the final laminate was not drastically affected. Furthermore, the maximum tensile load that could be applied without degrading the performance of the embedded energy cell was found to be about 50% of the ultimate tensile strength of the carbon fiber reinforced plastic (CFRP).

The manufacturability of the same all-solid-state thin-film battery was then systematically investigated by Gasco et al. [25]. This analysis focused on the effect of curing conditions on the market-available thin battery by taking into account temperature, pressure, and resin embedding. Four different values of temperature between 120 °C and 199 °C were considered, and the capacity retention was measured for the surviving batteries. The study demonstrated that temperature is the most influential process parameter on battery life and that embedding for this class of devices is suitable up to a maximum curing temperature of 132 °C. Three failure modes were observed, and the main limitations were the melting point of the thermoplastic sealant and the maximum operating temperature of the lithium anode, which was below 180 °C. The first one, which should ensure the electrical insulation and the sealing of the highly reactive cell components, melted at around 100 °C and tended to react with the lithium, creating dark circular patches on the anode (see Figure 6(e/f)). The other one created localized grey spots even below its melting point (see Figure 6(c/d)), and after that, it became black with a sudden loss of voltage (see Figure 6(g/h)).

Parallel studies corroborated by a detailed and systematic analysis of multifunctional composite structures were done by Thomas et al. [5] for marine applications. Three different configurations of sandwich panels were manufactured using high-performance carbon-glass composite laminates as skins, a closed-cell styrene foam as core, and an epoxy glue compatible with the lithium-ion cell. Kokam lithium rechargeable cells rated at 3.7 V with a specific capacity of 50 mA h/g were used as energy storage elements embedded in composite skins. A commercial plasma-etch treatment followed by dip-coating in epoxy resin was applied to the cell surfaces to improve the bonding strength. The prototypes were manufactured in multiple steps and the final structures were assembled using a low temperature curing adhesive to avoid any thermal stresses for the batteries. The configurations adopted in this investigation for engineering the unmanned underwater vehicle hulls are shown in Figure 7. The three-point bending test was chosen in this circumstance to evaluate the sandwich performance in terms of bending stiffness and maximum bending load, while the specific energy and the energy density were estimated with a series of constant current discharge tests after a full charge. A slight increase in the stiffness and a reduction of ≈60% in the flexural strength were found from the direct comparison of the sandwich panels. It is worth noting that all the tests revealed a common root cause of failure in the delamination in between the cell and the face sheets, showing a weak point for these configurations. From the electrical investigation, values of specific energy above 30.0 W h/kg were found for a discharge rate of 1 C in all the multifunctional configurations, with a peak of 45 W h/kg for the modular stiffener.

Moved by the encouraging results obtained by Thomas et al. [5], Shalouf and co- workers [38] performed a comparative study of the mechanical performance of sandwich panels with embedded lithium-ion batteries. The work covered the effect on electrical performance under different loading conditions and for different stress levels. Composite sandwich panels were manufactured with 199 gsm twill T300 fabrics face-sheets and with a Corecell A80 foam core bonded together with Araldite adhesive. Panels without the cells were produced as control specimens, and their mechanical performance compared to those with Kokam lithium polymer cells embeddings. No special precaution was taken to improve the bonding between the battery and the face-sheets. Specimens were tested under tension and three-point bending in compliance with the American Society for Testing and Materials (ASTM) standards D3039 [39] and D790 [40], while compressive tests were performed using an anti-buckling guide. Determining the ultimate loads, new samples were subjected to progressive load steps (25% of the maximum load for tensile and compressive tests and 20% of the ultimate flexural strength) and unloaded to monitor the degradation of the electrical performance produced by different levels of mechanical strain. The results highlighted a mild effect of the battery embedding on both the tensile stiffness and strength, while a consistent reduction in the flexural stiffness was observed. The compression tests, however, highlighted a critical aspect of the technological process for embedding the cell. Due to the battery layer-wise structure itself, the lack of any internal mechanical connection led to premature buckling phenomena observed in the embedding region, with a consistent reduction of the failure loads. Figure 8 shows the results obtained during this experimental campaign, highlighting the detrimental effect on the panel compressive properties in absence of improved adhesion between the cell layers and the face sheets.

Furthermore, this study underlined a linear relationship between the applied strain and the reduction in capacity change of the battery, as shown in Figure 9.

In a more recent follow-up study, Thomas et al. [41] investigated energy storage performance and the effect of the electromechanical coupling under sustained bending and hydrostatic pressure. The load relaxation under three-point bending tests was measured for the three structural batteries presented in Figure 7, and charge/discharge loops were performed with the sustained load. The results revealed a not negligible load relaxation for all the solutions, with a minimum of ≈6% for the structural laminate and a peak of ≈18% for the sandwich one, while the effect of electrical charge and discharge on the mechanical load was constantly lower than ≈2%. Hydrostatic tests at 10 and 20 bar, instead, proved a dependence of the electrical performance on this loading condition, with a reduction in the specific energy of ≈8% for the laminated and a modular structure that reduced to ≈4% of the sandwich one. This study shed some light for the first time on the main limitation of this emerging field: the fabrication of robust structural batteries. The maximum operating temperature of the battery has posed a limit to the mechanical performance of this type of power composite. The difficulties in co-curing the sub-components and the need for adhesives with a low curing temperature together with the absence of any transverse reinforcements and the poor shear resistance of traditional Li-ion batteries determined solely by the inner inter-layer friction and the packing deformation have posed additional challenges for the researchers.

A remarkable improvement in the flexural properties of composite structural batteries was obtained by Ladpli and coworkers [27]. A graphite nickel-manganese-cobalt (NMC) battery with a polyolefin separator was encapsulated in carbon fiber composite laminates with integrated polymer rivets used for stabilizing the battery layers. The battery core was perforated, creating circular holes in the layers before the lamination, while the face sheets were manufactured separately and barrier layers were bonded on the skins. The battery layers, the polymer rivets, and the composite laminates were compression molded at 100 °C and 0.5 MPa; the cells were then filled using a standard lithium-salt liquid electrolyte (LiPF_6_ in ethylene carbonate (EC)/ dimethyl carbonate (DMC)/ diethyl carbonate (DEC) organic solvent, 10 mL) before sealing. The final structural configuration is presented in Figure 10. Two different meshes of rivets were investigated, the 4 × 4 and the 5 × 5 ones and were compared with a configuration without rivets. The electrochemical performance of the new constructive solution was compared with a reference pouch cell. The initial discharge capacity of the cells was quantified during constant current cycling at C/10, while direct current impedance was evaluated every 10% of the depth of discharge. The quasi-static three-point bending test was used instead to determine the flexural stiffness and strength of the cells. Cyclic three-point bending was eventually performed at 20% of the static failure load and interrupted every 100 cycles to monitor the electrochemical performance degradation under fatigue.

From comparison with the reference pouch cell, an increase of ≈30% of the direct current impedance was observed and presumably induced by the high-temperature, high-pressure fabrication process. Moreover, a specific energy reduction of ≈40% was registered, while the volumetric one was reduced by up to ≈60%. The measured values of capacities were consistent with the expected theoretical ones, considering that a surface area reduction after the perforation and a capacity retention comparable with the pouch cell was observed. The beneficial effect of rivets has remarkable flexural stiffness comparable with the high-performance structural sandwich in which the entire battery volume was replaced by the polymeric core. The presence of rivets, in fact, enhanced the transfer of the shear stress and inhibited the slipping motion in between the electrode layers. Fatigue results confirmed this beneficial effect, showing a reduction of ≈3% in the energy storage composites capacity and an increase of ≈2–3% in direct current impedance after 1000 cycles. The panel without rivers had instead a drop of ≈25% in capacity and an increase of ≈21% in impedance for the same number of cycles. For sake of completeness, Figure 11 summarizes electrochemical, quasi-static, and fatigue results for this configuration.

More recently, an extensive experimental campaign on multifunctional composite laminates (see Figure 12a) and sandwich structures (see Figure 12b) for automotive applications with embedded lithium-ion polymer was conducted by Galos and coworkers [8,9,10,11,12]. The two constructive solutions adopted for this series of experiments had in common the use of LP423043 LiPo batteries supplied by LiPol Battery Co. Ltd. These 40 × 30 × 4 mm^3^, 10 g, 500 mA h, 3.7 V batteries were fabricated with lithium cobalt oxide and graphite electrodes and a N-Methyl-2-pyrrolidone (NMP) Polyvinylidene fluoride (PVDF)-based polymer electrolyte with lithium hexafluorophosphate salt (LiPF_6_). The T300 plain-weave carbon fabric with an areal density of 200 gsm and a low-temperature bisphenol-A-based epoxy system was adopted as structural elements. The multifunctional laminates were manufactured stacking 24 plies with a cross-ply [0/90] pattern; the 18 central plies were cut to allocate the batteries and the whole assembly was cured at room temperature. The sandwich panels, however, were assembled using a Divinycell H100 PVC foam core with a density of 100 kg/m^3^ that was ad hoc cut to house the batteries. In contrast to what has been presented up to this point, these sandwich composites were co-cured at room temperature without extra pressure in order to avoid any damage to the batteries. It is worth noting that each battery was surrounded by a non-negligible resin-rich zone. A sensitivity study of the integrated number of batteries and of their position was performed for different loading conditions.

In their first work, Galos and coworkers [8] focused on the vibrational and acoustic properties of the panels for optimizing EV applications. Beam-shaped specimens were excited via a mechanical shaker, and the first three natural frequencies of the bending modes of vibration were measured, as well as the damping ratios. The presence of batteries in laminated configurations resulted in a reduction of the first vibration mode proportional to the number of batteries, while the first damping ratio remained almost stable; no clear correlation was detected for the other modes. Sandwich panels, as well, showed a gradual but less pronounced decrease in the first vibration mode induced by the increasing number of batteries. The following vibration modes, however, did not show any relevant change.

The same group [12] conducted parallel research on the effect of embedded batteries in the tensile properties of both composite laminates and sandwich batteries. Tensile tests were conducted in compliance with ASTM D3039 [39]; experiments were compared with finite element (FE) models, and a sensitivity study on the number of embedded batteries was carried out comparing the results with composite structures lacking any embedded batteries. The results for laminated batteries highlighted a progressive reduction in stiffness with the increase of the number of batteries from 18% with one for a nominal specific energy of 12.1 W h/kg up to 45% with three batteries with a spacing of 28 mm and a specific energy of 57.6 W h/kg. The reduction in tensile strength, however, reported a reduction of ≈60% for all the embedded laminates independent of the number of embedded batteries. Different was the case of the sandwich batteries, where replacing the PVC core with batteries had no effect either on the tensile stiffness of the panel or on its tensile strength and produced only an increase on the nominal specific energy of the resulting engineered material. Cyclic charge/discharge tests were also performed at 1 C during the tensile tests at approximatively 30% and 60% of the failure stress. No degradation induced by the tensile strain was noticed for the two types of configurations investigated.

Recent investigations [10] highlighted the flexural properties of sandwich panels with embedded batteries. A sensitivity study on the effect of the number of batteries and battery spacing was carried out via three-point bending tests in compliance with ASTM D790 [40]. Two specific energy, 43.3 W h/kg and 71.7 W h/kg, and different span-to-thickness ratios, 15-to-1 and 33-to-1, were investigated, finding two dissimilar failure modes: the plastic indentation of the core for short samples and the core cracking for the longer ones (see Figure 13). Electrical performance was monitored during and after the three-point bending execution with a charge/discharge test at a constant current rate of 1 C.

It is worth noting that the charge/discharge capacity did not change significantly for the whole execution of the test, revealing a good capacity of the LiPo battery used in this study to held flexural deformation even in elasto-plastic regime. Reduction of ≈25% in specific stiffness and of ≈15% in specific strength were instead recorded for single battery specimens, and the presence of more than one battery led to a remarkable reduction in specific mechanical properties.

In more recent work, Attar et al. [9] studied the effect of the same LiPo pack embedding in the compressive behavior of laminated batteries under quasi-static and fatigue conditions. Following the same scheme, previously adopted for the tensile and the flexural specimens, the sensitivity to the number and the disposition of the batteries in the CFRP plates was investigated according to the ASTM standard [42]. Twenty-four ply 100 mm × 150 mm specimens were cut to obtain slots for embedding the batteries, and the specimens were cured out of the autoclave and at room temperature. Quasi-static tests revealed a reduction of 18% in the compression modulus and a reduction of 43% in compressive strength for the single battery specimen, with the long side of the battery aligned with the loading axes. A dependence of the mechanical properties proportional to the cross-sectional area of the specimens occupied by the batteries was also identified and is shown in Figure 14a. The outcome of the fatigue test, presented in Figure 14b, revealed a degradation of the mechanical performance comparable to the one observed under quasi-static conditions. Electrical performance, as seen for the tensile and the flexural tests, reported no degradation under loading, but several connection tabs to the battery electrodes failed under fatigue loading.

Confirmation of the high capabilities of LiPo batteries to hold extreme deformation was demonstrated by Galos et al. [11] in a recent work. Analogously to what was done by Saharei et al. [43,44] for the market-available Li-ion cells, Galos et al. [11] carried out an extensive experimental campaign on the mechanical properties. The shear behavior was characterized via the hole-punch test, as described in ASTM D732 [45]; the flexural stiffness and strength were measured via the three-point bending test, while the compressive properties were measured via the NASA short block test [46]. LiPo batteries showed high electrical performance retention under all of these different loading conditions and a good capability to accumulate plastic deformation without degradation.

More recently, Pattarakunnan et al. [47] showed the results of an impact test campaign performed on laminate and sandwich composite structures, including LP423043 batteries. In compliance with ASTM-D7136 [48], 150 × 100 × 5.5 mm^3^ samples with a single centered battery cell were impacted with a drop weight at different energy levels ranging from 2 J up to a maximum of 8 J. The presence of the battery for the CFRP laminates was the cause of the damage initiation and propagation, producing a significant increase in the energy-absorbing capabilities of the structure. In the case of the sandwich panels, however, the batteries changed the failure mode of the structural elements but did not produce any remarkable change in the amount of energy absorbed during the impact. In particular, embedded samples were more prone to fail due to delaminations between the battery and the face sheets, while monofunctional panels highlighted a failure initiation on the impacted skin. Measurements of the internal capacity and resistance of the battery after the impact revealed that the LiPo batteries were capable of completely preserving their properties until the final failure that, for the investigated structures, was for impacts with an energy above 6 J for both the investigated configurations.

Notwithstanding, a big step forward has been done in improving the mechanical be- haviour of embedded structural batteries by introducing transverse reinforcements [27] or adopting co-curing as a manufacturing option [8]; still, much of the margin in optimizing this process has to be exploited. The study of new interfacial treatments for both the batteries and the face sheets as well as the use of alternative solid-state batteries with enhanced mechanical properties and the increase in the operating temperature of the batteries may contribute positively to the development of the structural batteries of the future.

## 4. Multifunctional Materials

This section describes a different concept for manufacturing structural power composites, the one defined by Thomas et al. [30] as ‘multifunctional materials’. The main goal in developing multifunctional power composites is to create materials where the main components of the battery are also primary load-carrying elements, or the other way around, the structural constituents of the composites work together as a solid battery.

This idea dates back to 1999 when Chung and Wang [49] introduced for the first time the idea of using carbon fibers to make electronic devices. This step paved the way to the concept of structural capacitors, where the material itself is capable, at the same time, to bear the mechanical load and to store electric power. Starting from this work, in 2001 Luo and Chung [50] manufactured structural capacitors with a carbon fiber epoxy matrix and a paper interlayer. Another preliminary experimental work on multifunctional composites for energy storage applications was carried out in 2006 by Wetzel et al. [51]. With this exploratory activity, the concepts of layered multifunctional structural capacitors, batteries and fuel cells were investigated. These three new products were designed and prototypes manufactured with the purpose of creating a new material where the electrical parts were also load-carrying elements.

A first milestone in the design and fabrication of multifunctional lithium-ion batteries was placed by Liu et al. [52], who realized carbon nanofibers structural batteries with tunable mechanical properties. The devices were realized with PVDF-based fiber reinforced composite with different fillers as the active material. Graphite was used as the anode, while lithium cobalt oxide, LiCoO_2_, was the cathode. The separator, however, was manufactured using a polymer blend of PVDF-HFP (hexafluoropropene) with an electrolyte solution of 1M LiPF_6_. Samples of anodes, cathodes, and separators were then bonded to copper and aluminum grids by hot pressing at 150 °C to assemble the final batteries. The novel design showed good mechanical properties with a tensile modulus of 3.1 GPa but a modest specific energy of 35 W h/kg*_Str_* at a discharge rate of C/20 caused by the poor ionic conductivity of ≈10^−5^ S/cm of the structural electrolyte.

Starting from these works, in 2010 Eksted et al. [15] developed a laminated structural battery manufactured with carbon fiber, an aluminum mesh, and a glass fibers separator. The investigation was carried out with two different types of electrolytes, a gel and a polymeric one, with the addition of LiPF_6_ salts. The outcome of the activity was a working proof of concept of structural batteries with an open cell potential of 3.3 V that paved the way for a new field of research.

Although the battery performances were not impressive from an electrical point of view, mainly due to the low ionic conductivity of the structural electrolyte, this work provided the cue for a further step. A few years later, in fact, Asp and coworkers presented and patented [24,53,54] an innovative concept of structural batteries based on the idea of single fiber electrodes. This conceptual design, addressed as the 3D-fiber structural battery, was originally built from approximately one thousand carbon fibers electrochemically coated with a solid polymer electrolyte (SPE) and embedded in a cathode-doped matrix material. The concept was introduced for the first time by Leijonmarck et al. [55] who coated tows of unsized IMS65 carbon fibers with two different types of SPEs. After a preliminary screening based on the process and the constituent parameters, methoxy polyethylene glycol (350) monomethacrylate (SR550) monomer dissolved in the desired concentration of dimethylformamide (DMF) and lithium trifluoromethanesulfonate (Li-triflate) were used to coat the fibers of the composite battery prototype. These coated fibers were then used as the negative electrode in a Li-ion battery with lithium metal as counter electrode and a liquid electrolyte showing a specific capacity of 107 mA h/g for a current rate of 1 C. Moreover, SEM images showed a uniform and hole-free thin coating that allowed reducing the lithium transport distance, typically of 20–25 μm in traditional Li-ion batteries, down to 500 nm, compensating in this way the scarce conductivity that characterizes the SPEs.

In light of these initial activities, two technologically appealing configurations for structural batteries came to light: the laminated one shown in Figure 15a and the 3D one displayed in Figure 15b.

The recent developments in the field of multifunctional power composites are presented in the following sections, focusing on the main components that are present in all batteries: the anodes, the cathodes, and the electrolyte. More in detail, it is shown how traditional structural elements of advanced carbon fiber reinforced plastic such as the fibers, the matrix, and the interfaces have been rethought and tuned to produce new energy storage devices.

### 4.1. Structural Anode

Due to the affinity with the graphite commonly used in lithium-ion batteries, carbon fibers were immediately identified as a possible candidate for electrodes in structural storage materials [56]. Their good electrical conductivity and the high specific mechanical properties, together with a carbonaceous microstructure that promotes good lithium-ion intercalation, pushed several research groups to investigate the fibers’ capabilities for structural power composites.

The first systematic studies on carbon fibers were conducted by Kjell and coworkers [57], who investigated several grades of commercially available polyacrylonitrile (PAN)-based carbon fibers for structural lithium-ion composite batteries. The research focused on understanding how the lithiation rates and the number of fibers per tow affected the capacity of the fibers when used as both current collector and negative electrode. Moreover, the effect of sizing and of fiber modulus and strength were analyzed to evaluate the effect on fiber electrochemical performance. The electrochemical capacity for the different carbon fibers was measured using galvanostatic cycling between 0–1.5 V vs. Li/Li^+^ with a current corresponding to 100 mA/g. A large variability of the electrochemical capacity between different grades of fibers was found. Maximums of 177 mA h/g were noticed after 10 cycles for unsized intermediate modulus fibers, while very low values below 30 mA h/g were identified in high modulus tows. Furthermore, the effect of different lithiation rates spanning from 1 C to C/10 was investigated for the two highest capacity fiber systems. The results revealed that, as expected, the higher the lithiation rate was, the lower the fiber capacity became, with a doubling of the capacity when the current rate was reduced by a tenth. In the following studies, Jacques et al. [58,59] investigated the effect on the tensile properties of the fibers induced by the lithium-ion insertion and extraction into the microstructure. Due to their higher electrochemical capacity, the study focused only on continuous intermediate PAN-based carbon fibers; more specifically, the Toho Tenax IMS65 and the Toray T800 were selected for their specific capacity of about 135 mA h/g and the excellent capacity retention after 10 cycles. Fiber yarns were split to obtain 22 mm long specimens; the samples were pre-stretched and tabbed on the ends to get a uniform load transfer and to minimize the risk of stray fibers to induce short-circuits. Samples were placed in a glove box and used as positive current collectors in layered electrochemical pouch cells with a lithium metal foil as the counter electrode and a glass microfiber filter impregnated with liquid electrolyte as the separator. Further details of the materials used for these studies are reported in Table 2. It is noteworthy that in this specific test set-up the fibers had a higher standard electrode potential than the lithium metal and were therefore the positive electrode of the cell.

Current collectors were then connected, and the cell enclosed in a laminated bag and the vacuum was drawn from the bag to guarantee good contact between the layers. Figure 16 shows the tensile specimen and the cell used by Jacques et al. for their investigations. Tensile tests were carried out after a specific number of lithiation and delithiation cycles for both the investigated fibers. The results showed that neither the lithium intercalation nor the electrochemical cycling affects the fiber stiffness, but, after the first lithiation, the fibers’ strength had a 20% drop that was partially recovered after a delithiation phase. The repetition of electrochemical cycles, however, did not produce any further degradation. This effect is related to the fact that some ions remain trapped after the first intercalation in the carbon fiber microstructure, producing a solid-electrolyte interphase (SEI) that induces an irreversible loss in both carbon fiber mechanical strength and cell capacity.

Further investigations carried out by Jacques et al. [60] studied the effect of lithium-ion intercalation in carbon fiber expansion. This is a crucial aspect for the application of this class of multifunctional electrodes in structural power composites since it could produce internal stresses and strains that have to be taken into account during the design phase. The longitudinal expansion was measured via a tensile test on electrochemical cells like the one presented in Figure 16. Samples were pre-tensioned and subjected to electrochemical cycling at different current rates. Considering the carbon fiber stiffness stable during the testing strain range, variations in force were associated with fiber expansion and contraction during cycling. More in detail, during the first cycle a permanent axial extension and a reduction in the capacity loss were recorded, induced by the formation of the solid electrolyte interphase. Subsequent cycles were characterized by a force drop during the lithiation phase, indicating a fiber expansion and a recovery to the original value at the end of the delithiation. The investigated current rates of 6 C, 3 C, 1 C, 0.4 C, and 0.1 C revealed what was expected: that the higher capacity loss was recorded for high current rates while for current rates below 1 C the final capacity stabilized. In contrast, regarding the mechanical properties, it was noticed that the higher the current rate were, the lower the axial deformation became, indicating that the fiber expansion increased with the amount of lithium intercalated during the lithiation. Peaks of 1% in axial elongation were recorded in both the studied fiber systems at the lowest charging rate of 0.1 C, with values of capacity close to the theoretical 372 mA h/g. The transverse expansion was also determined using scanning electron microscopy, detecting a radial expansion between 8% and 13% for fully lithiated fibers and a permanent residual deformation of 2–3% when in the delithiated condition.

Further confirmation of the good properties of carbon fibers as electrodes for structural power applications was provided by Hagberg et al. [61]. A comprehensive analysis of PAN-based carbon fiber systems was done via accurate galvanostatic cycling and confocal Raman spectroscopy. The Coulombic efficiency together with the fiber capacity were measured with lithiation/delithiation cycles between 0.002 V and 1.5 V vs Li/Li^+^ at three different current rates: C/10, C/20, and C/50. A test set-up analogous to the one previously described was also used in this work and the details are reported in Table 2. Electric performances were compared with a commercial graphite-based micro beads electrode (Qualion MesoCarbon MicroBead (MCMB)) subject to the same cycling sequences. Raman spectroscopy, however, was used for studying the fibers’ microstructure: in particular, the amount of disordered carbon and the graphitic content. The results obtained from this analysis are shown in Figure 17, where the delithiation capacity is presented as a function of the number of loading/unloading cycles, and the Coulombic efficiency is plotted as a function of the measured fiber capacity.

After the first discharge, the capacity reached values between 200 and 250 mA h/g for most of the intermediate modulus PAN-based fibers, while the ultra-high modulus fiber showed a lower capacity of ≈150 mA h/g. It is worth noting that the capacity of the unsized IMS65 system started at ≈350 mA h/g, as high as the commercial MCMB electrodes, and dropped to values lower than 300 mA h/g after 30 cycles. Coulombic efficiency was excellent for all the investigated PAN-based fibers, close to 100% after the tenth cycle for the faster discharge rate, even exceeding the MCMB. This behavior was attributed to the small surface area of the carbon fibers compared with the MCMB that led to a small and stable solid electrolyte interface that could guarantee a very high battery lifetime. Raman spectroscopy results revealed that PAN-based fibers have a disordered and amorphous structure with nanocrystals that seem to promote the Li-ion intercalation.

Further light on the topic was shed by Fredi and coworkers [64], who used high- resolution transmission electron microscopy (HR-TEM) and in situ Raman spectroscopy to comprehend the connection between the carbon fiber microstructure and the lithium-ion intercalation mechanism. Fibers with a different turbostratic graphitic microstructure compared the intermediate modulus high-capacity IMS65 and T800 and the high modulus low-capacity M60J. HR-TEM revealed that the microstructure of the M60J fiber had an ordered homogeneous microstructure composed of relatively large crystals (>300 Å) on the longitudinal direction stacked at >100 Å in the thickness, while the structure of the intermediate modulus fibers were highly disordered and composed of very small crystals that were a characteristic length smaller than 30 Å. This disordered microstructure is the reason why intermediate modulus fibers have additional sites for ions deposition that produce an increased capacity, while a more ordered configuration, such as one of high modulus fiber, induces an intercalation mechanism closer to that of graphite but disturbed by the presence of a turbostratic disorder that strongly affects the resulting material capacity.

All these characteristics confirmed the applicability of intermediate modulus carbon fibers as an anode for the multifunctional composite materials of the future.

More recently, a wide experimental activity on carbon fibers coated with structural electrolytes was performed by Schutzeichel et al. [65]. The work focused on the study of the multiphysical characterization of the state-of-the-art polymer-coated carbon fibers, including electro-thermomechanical properties within a range of temperatures relevant to the aircraft industry. Unsized IMS65 carbon fiber bundles were coated with a copolymer obtained with a 1:1 combination of SR550 (high conductivity, low stiffness) and SR209 (low conductivity, high stiffness) polymer with 8% of lithium salt. SEM images, taken before the testing, verified the presence of a homogeneous thin layer of polymer electrolyte around the fibers. Measurements of the specific longitudinal resistance of the unsized IMS65 fiber confirmed a value of 1.35 × 10^−3^ Ω cm, and the coating showed no effect on this value, and no variation with temperature was identified. The static Young’s modulus of the composite was found to be reduced by 15% compared with the bare fibers, and a reduction in the range of 20% to 30% of the storage modulus was recorded in the temperature range of −80 °C to 130 °C. This result not only provided relevant physical properties for multifunctional composites but gave a relevant insight into the need for multiphysical analyses for the study of structural power solutions.

Given the promising results obtained with the use of carbon fibers as a structural anode in lithium-ion solutions, Harnden and coworkers started an investigation of the electrochemical and mechanical properties of carbon fiber intercalated with sodium [66] and potassium [67]. Recently, concerns about lithium resource shortages have led to an interest in potassium-ion and sodium-ion chemistry in battery applications. Moreover, the bigger size of potassium and sodium atoms compared with lithium suggested that the use of these materials could produce greater expansion during ion intercalation, making these solutions extremely appealing for actuators. T800H carbon fibers have been used in these investigations to manufacture pouch cells used for the experimental campaign. In the sodium-based samples, a sodium metal foil was used as counter electrode, and the separator was impregnated with a 0.6 M NaPF6 in diglyme solution. In the potassium cells, however, a potassium metal foil was used as a counter electrode, and the ionic liquid was realized with 0.5 M KPF6 in propylene carbonate (PC). All the cells were cycled in a potential range from 2.5 V to 0.01 V vs X/X^+^, while the mechanical properties were investigated using a micro-tensile tester. As per the lithiation case, the cycled fibers’ stiffness did not change during sodiation and potassiation cycles, but the ultimate tensile strength dropped by 27% and 12%, respectively, at the end of the first intercalation. A recovery of this value was recorded at the end of the deintercalation with a final strength reduction of 6% in both cell types. Sodiated specimens with a current rate of C/3 recorded a stable specific capacity at around 90 mA h/g after the first cycle, with a first cycle drop of 30 mA h/g and an irreversible axial expansion of ≈0.06% due to the SEI formation. The analysis of the pouched cells with potassium ions revealed for an analogous current rate a first specific capacity of 133 mA h/g that stabilized only after ≈20 cycles to 40 mA h/g. Measurements of the voltage-strain coupling showed a bilinear behavior for the sodiated fibers, with a maximum coupling factor of 0.15 V/unit strain recorded for a state of charge of 45%. The same parameter in the potassium cells has a peak of 0.25 for a state of charge (SOC) of approximatively 50%; 7 times less than the lithiated carbon fibers [68]. Compared with the sodiated solution that had an actuation energy of 65 J/kg, the potassium cells at a similar current rate had a slightly higher value of 117 J/kg but still not comparable with the 1600 J/kg recorded for lithiated cells. Notwithstanding that potassium- and sodium-insertion in commercial PAN-based fibers provide an alternative solution for expanding the carbon fiber systems functionality, the lithium-insertion is still the most promising way for high-performance structural batteries. The first steps have been taken in the direction of using the carbon fibers as a structural anode for multifunctional composites, but still many aspects, including dendrite formation [69,70] and crack nucleation during the lithiation/delithiation cycles, have to be deeply addressed.

Alternative techniques to increase carbon fiber electrochemical performance such as the one based on hierarchical network architecture already developed for carbon fiber paper [71] and 3D carbon nanofibrous network [72,73] remain unexplored research fields for structural relevant carbon fibers.

### 4.2. Structural Cathode

The promising results on the application of PAN-based fibers as anodes for power composites has recently pushed a small number of research groups to investigate the possibility of manufacturing carbon fiber-based cathode current collectors. The first attempt was carried out by Hagberg et al. [62] who used an electrophoretic deposition (EPD) technique to deposit active electrode material on carbon fiber substrate. In this work, unsized Hexcel AS4 PAN-based fibers were covered with carbon black (CB)-coated LiFePO_4_ (LFP) particles with a specific capacity of 150 mA h/g. PVDF was used as binder and different mixtures of LFP, CB, and PVDF were investigated. Scanning electron microscopy (SEM) analyses were performed to investigate the quality and the composition of the deposition. A pouch cell with lithium foil as the negative electrode, a glass microfiber filter as separator, and liquid electrolyte was used to perform the electrochemical characterization. More details about the test set-up are reported in Table 2. Coulombic efficiency and specific capacity were measured with galvanostatic cycles between 2.8 and 3.8 V vs Li/Li^+^ for different current rates. Mechanical tests such as the three-point bending and the double cantilever beam tests were done to investigate the interphase composite dominated properties such as the transverse modulus, the transverse ultimate strength, and the mode I interlaminar fracture toughness. SEM images showed that the EPD produced a variable thickness porous coating characterized by a well-dispersed presence of LFP particles. The average specific capacity at 0.1 C was between 62 and 108 mA h/g for the different LFP:CB:PVDF compositions. A sensitivity analysis to the amount of PVDF revealed that values higher than 4% drastically reduced the capacity retention for current rates higher than 0.1 C, while values of 6 % of CB showed better performance. A comparative study of the mechanical performance of LFP-coated fibers and uncoated fiber-based composites revealed a high adhesion. No reduction was measured for both the transverse modulus and ultimate transverse strength. Double-cantilever beam (DCB) tests showed a comparable value of the mode I interlaminar fracture toughness between the coated and the uncoated fibers, confirmed by optical microscopy of the fracture surface that revealed a good impregnation of LFP-coated tows.

An alternative technique for realizing carbon fiber-based positive electrodes was introduced by Bouton et al. [74]. In this work, a layer-by-layer deposition process of LiFePO_4_ on carbon fiber was presented. The approach, via a carbonization step, ensured a transformation of the organic and insulating binders in an electrically conductive network and removed the possibility of chemical side reactions between the electrolyte and the binder. Pouch cells with layer-by-layer positive electrodes, realized with two different LFP solutions and two different carbonization temperatures, were manufactured and galvanostatically tested with a discharge rate of 0.1 C. The results showed values of the specific capacity of 100 mA h/g.

In a more recent study, Moyer and coworkers [29] combined lithium-ion active materials with carbon fiber tissues to realize pouch-free laminated energy storage composites. Lithium iron phosphate incorporated with carbon nano-tubes and graphite were coated onto 60 × 60 mm^2^ thermally processed carbon fiber weave material to produce the cathode and the anode of a full structural battery. The active layers were then divided by a separator where a liquid electrolyte was infiltrated to allow ion transportation, and the layup was then sandwiched with 84 × 84 mm^2^ carbon fiber laminates and cured, as shown in Figure 18a.

The battery-specific energy was determined by galvanostatic cycles at different current rates, showing a capacity of ≈30 mA h/g at 0.1 C and a specific energy of 36 W h/kg. Due to the need for a structural battery to keep electrical performance even under mechanical load, the electrochemical tests were repeated at three different levels of tensile stress: ≈50% and ≈95% of the ultimate tensile strength. This revealed a progressive reduction of the cell capacity with the increase of the loading condition, which was attributed to the shear stress-induced delamination at the interfaces. The prototype of the battery was also integrated into a CubeSat structure to show how structural power composites could be used to provide an integrated power delivery system, saving weight and volume. In follow-up work, [63] Moyer and coworkers improved both the mechanical and electrochemical properties of the batteries via a polyacrylonitrile (PAN) coating of the electrodes (see Figure 18b). The PAN coating, commonly used for improving the mechanical performance of carbon fibers in lightweight structures, was used to simply sandwich the active battery material to the carbon fibers for both the electrodes. A Celgard 2525 separator was soaked with 1 M LiPF_6_ in diethyl carbonate 1:1 and interposed in-between the electrodes to create a lithium-ion battery that was encapsulated in carbon fiber laminates. A comparative study between PAN-coated and un-coated electrodes was performed to show the mechanical and electrochemical importance of engineering interfaces in energy storage devices. Mechanical analyses of the interface adhesion have been carried out via the lap-shear test of the electrodes, only highlighting an increase in the shear strength of 40% for the anode and up to 80% for the cathode. After 100 galvanostatic cycles at a current rate of C/10, the coated cell showed a specific capacity of 20 mA h/g with a retention of 80%, while the non-coated counterpart had a retention inferior to 65% with a specific capacity of ≈10 mA h/g. The PAN-based structural battery demonstrated a specific energy of 52 W h/kg approximatively more than two times higher than the uncoated one and one-third of commercially available packaged Li-ion cells. In addition, electrochemical impedance spectroscopy (EIS) tests indicated an initial resistance of 60 Ω that increased to 240 Ω after 100 cycles, showing a much better stability than the battery without the coating that initiated cycling, with a resistance of 45 Ω that increased until 1140 Ω. All these evidences confirmed the beneficial effect of PAN coating that increases the active material adhesion and reduces the nucleation and the propagation of delaminations on the electrode interfaces during the SEI formation and the subsequent electrochemical cycling. Overall, this study showed the relevance of solid interfaces as an essential step to realize storage energy composites with enhanced multifunctional properties. The details of the materials used for the manufacturing of this battery are reported in Table 2.

### 4.3. Solid Polymer Electrolytes

Most of the previously presented innovative battery configurations and pouched cells used for evaluating the electrochemical performance of both anodes and cathodes were realized with liquid electrolytes. These liquids are those generally used in the market-available lithium-ion batteries due to their high ionic conductivity, which can reach values up to ≈10 mS/cm. Unfortunately, their introduction in structural power composites is detrimental from the mechanical point of view, which makes them less appealing for structural power applications. For this reason, the realization of solid electrolytes with high conductivity has been and will be the keystone for the fabrication of robust structural batteries. A solid structural electrolyte is a material that is responsible at the same time for the conduction of the lithium ion and for the load transfer capabilities of the whole multifunctional solid. To be effective, these electrolytes should be capable of providing a compressive and shear stiffness between 0.1 and 1 GPa, and of guaranteeing an ion conductivity on the order of ≈1 mS/cm, providing a good mechanical and electrical connection with the other battery components. Exploratory studies were initially done for manufacturing solid electrolytes but due to their poor ionic conductivity they were recently supplanted by bicontinuous engineered solutions. In this section, the recent history of structural polymer electrolytes is traced, focusing on the processes and the materials involved in the evolution of these critical components.

Preliminary investigations were carried out by Snyder and coworkers [75,76] who studied the potential use of polyethylene glycol (PEG)-based vinyl ester polymer electrolyte for multifunctional applications. Several conductive resins were manufactured, and their conductivity measured; mechanical properties were, however, evaluated via compression tests and dynamic mechanical analyses. As shown in Figure 19, the results conducted on a relevant number of solid polymer electrolytes showed a maximum achievable value for the ionic conductivity below 10^−4^ S/cm and with a compressive stiffness lower than 10 MPa. Moreover, this investigation revealed a clear negative logarithmic trade-off between ionic conductivity and the compressive modulus, showing how any improvement in mechanical performance leads to a detrimental drop in ionic conductivity and vice versa.

A valid alternative to full solid electrolytes came from the work of Ji et al. [77] and Matsumoto et al. [78] who moved to a bi-phasic electrolyte configuration where one phase satisfied the mechanical requirement and the other maximised the ion conductivity.

Beginning from this concept, it was within the framework of the “STORAGE’’ projects that the idea of bicontinuous electrolytes took hold. In fact, Shirshova et al. [16], starting from a comparative study of composite supercapacitors, identified the bicontinuous electrolyte as a valid option to traditional gel. The investigation compared a PAN gel electrolyte and a solid one obtained via a multifunctional resin based on cross-linked PEDGE (polyethylene glycol diglycidyl ether). The resin was doped with LiTFSI, a hydrophilic salt commonly used in Li-ion batteries, while an ionic liquid (IL) was introduced into the matrix formulation to improve the salt solubility. By using these electrolytes, multifunctional supercapacitors were manufactured with electrodes of activated and non-activated plain-weave carbon fiber laminae and a glass fiber separator. From the analysis of both the mechanical compressive tests and the charge/discharge experiments, modest performances were observed but, nevertheless, the study successfully demonstrated the potential of these new materials. From this starting point, Shirshova et al. [79] performed a systematic analysis of bicontinuous liquid-epoxy systems to detect both an optimal material combination and a process for maximizing at the same time the ionic conductivity and the mechanical performance. The research covered the use of three different high-performance epoxy systems, the MVR444, the MTM57, and the VTM266, which were mixed with LiTFSI salts dissolved in the EMIM-TFSI ionic liquid in different ratios. The electrolytes were manufactured incorporating the components and curing them as prescribed for each resin system. Values of conductivity, glass transition temperature, and elastic modulus were compared. Promising multifunctional performances were measured for epoxy systems with a resin content between 30% and 50%. Values below 30% highlighted a very high resin fragility, while the system with more than 50% of the resin phase showed poor ionic conductivity due to the fact that the liquid phase was trapped into the solid. For the sake of completeness, the relevant properties of the noteworthy configurations are reported in Table 3, while the pictures of the MTM57/50% hierarchical microstructure with connected spherical nodules are shown in Figure 20.

In light of these results, the same research group [80] identified the MTM57-based electrolytes with a resin content of 50% as the most promising candidate for further investigation. A dedicated sensitivity study of this bi-phasic polymer to lithium salt concentration ranging from 0.5 mol/L up to 4.6 mol/L was carried out. SEM micrographs, such as the one shown in Figure 21 for the SPE with the highest lithium salt concentration, showed as an increase in the salt concentration, promoted a finer microstructure and a macroscopically more homogeneous material.

Unexpectedly, this trend showed how a higher concentration of lithium salt improves the robustness of the polymer but, at the same time, degraded the ionic conductivity, with variation in the range of 2 orders of magnitude. This reduction in ionic conductivity was attributed to various factors, including swelling of the epoxy chains by the electrolyte, the reactions of the cyclic carbonate with the primary amine of the formulation leading to the formation of hydroxyl-urethane groups, or the decreased microstructural length scale. The second part of the work focused on the analysis of the effect on the solid polymer electrolyte of different amounts of polypropylene carbonate (PPC). Values ranging from 0.15 g up to 15 g were investigated. As shown in Table 3, the addition of this organic solvent with a high dielectric constant and a good electrochemical stability aimed to increase the solubility of the MTM57 resin in the ionic liquid electrolyte. The final outcome showed that a high amount of PPC produces the same effect of a high concentration of lithium salt, thus promoting a more homogeneous structure and a degradation in the electrochemical properties.

The final act of this project reported by Greenhalgh et al. [17] included the design, manufacturing, and testing of a multifunctional supercapacitor prototype. The device was realized using activated T300 twill fabric as electrodes, the doped bicontinuous MTM57/50% polymer with different LiTFSI concentration as structural electrolyte, and a plain-weave glass fiber fabric as separator; the effect of carbon nanotubes fiber sizing was also investigated. Voltage chronoamperometry between 0 V and 1 V was used for the electrical characterization of the devices, while the in-plane shear modulus and strength of the supercapacitor were investigated using ±45° tensile coupons, and the compressive properties were measured in compliance with ISO 14126 [86]. The results provided a low value of capacitance compared with a conventional supercapacitor and sub-optimal mechanical properties in the matrix-dominated failure modes. However, albeit these limitations, the study provided the route for the following studies on structural power composites.

In 2016 Yu and coworkers [81] presented a methodology to improve the bicontinuous electrolytes by adding different contents of organically modified silicates (OLS). The study focused on the effect of the OLS on an epoxy resin system mixed with liquid electrolyte (EMIM-Tf_2_N + 1.3 g of PC) and LiTf_2_N lithium salt. This work, as shown in Figure 22, highlighted how the introduction of OLS can modify the epoxy/liquid microstructure from an interconnected series of epoxy flakes with discrete epoxy spheres in absence of OLS to a micro-morphology characterized by ridges and holes.

The results, including the dynamical mechanical analysis to determine the glass transition temperature *Tg*, the tensile tests in compliance with ASTM-790 [40] for the mechanical properties, and the electrochemical impedance spectroscopy for the ionic conductivity, are summarized in Table 3. The study proved how the OLS can be used to tune the electrical and mechanical properties of a bicontinuous electrolyte, showing a huge room for improvement for this kind of technologies. The same research group in 2017 [82] manufactured a structural negative electrode with carbon fibers and an SPE realized with diglycidyl ether of bisphenol A (DGEBA) epoxy resin E51 doped with a liquid electrolyte. The latter was realized with bis (trifluoromethane) sulfonimide lithium salt (LiTf_2_N) dissolved in a mixture of ionic liquid and 1 wt% of PC at the concentration of 2.3 mol/L. Mechanical and electrochemical properties of the electrolytes obtained with four different amounts of ionic liquid were independently measured, and the most relevant parameters are recorded in Table 3. The structural electrolytes were cured with T700S carbon fiber fabric without utilizing any separator, and the carbon fibers took the place of both the reinforcements and the anode. The electrodes were manufactured using the vacuum-assisted resin infusion process, and coupons were thermally cured. A sensitivity study on the amount of liquid phase was performed, measuring both the mechanical and the electrochemical properties of the final material. Results showed an optimal multifunctional electrolyte with a Young’s modulus of 200 MPa and an ionic conductivity of 0.1 mS/cm that led to a final longitudinal stiffness of 195 GPa for the whole composite. Eventually, the manufactured power composites were set up in a 2032 lithium coin cell in contact with the lithium metal, and the discharge capacity was measured. The best performing cell was achieved with an epoxy liquid ratio of 50%. This cell showed, however, a poor cycling stability, and its specific capacity was just 25 mA h/g, about one-tenth of the traditional liquid batteries with the same electrodes. In a more recent study, Zhao et al. [87] used the same technique to assemble laminated structural batteries with a layup configuration. A biphasic electrolyte based on EMIM-TFSI and LiTFSI salts was used to impregnate a 0.23 mm plain-weave T300 fabric used as the anode. The rest of the structural cell was manufactured using a LiFePO_4_-coated metal foil as the cathode and a polypropylene separator. Glass plain-weave reinforced epoxy pre-pregs were used for encapsulating the cell, and a sensitivity study of the number of CFRP layers was performed. The best values of charge/discharge capacity were highlighted for the laminate with two layers of unidirectional CFRP in which values of 26.8 and 7.6 mA h/g were measured, respectively, for the first cycle. Unfortunately, with this configuration, a continuous reduction of the composite capacities was found, highlighting a lack of cyclic stability and requiring design improvements.

Starting from the consideration demonstrated by Torquato et al. [88], that the optimal microstructure for multi-modal transport is the one with segregated phases, Gienger et al. [83] tried to improve the structural battery electrolytes (SBEs) multifunctionality via a complete separation of the two phases, creating a robust solid microstructure and backfilling it with high conductivity ionic liquid. Three different solid electrolytes were manufactured and compared. In this study, two multifunctional systems were produced by mixing EPON 828 and PACM resins with two different electrolytes, a 1 M LiTFSI in PC (σ = 5.10 mS/cm) and a 1M LiTFSI in PEG (σ = 0.88 mS/cm), with different mixing ratios. The third one was instead realized by curing the resin with a soluble substance that separated the phases during curing; this element was then removed and replaced by a 1M LiTFSI in PC liquid electrolyte. SEM images for the three multifunctional polymers for the highest multifunctional condition obtained for an electrolyte content of 65% are shown in Figure 23. From these pictures, it appears that the addition of 1 M LiTFSI in PC does not produce any morphological change in the resin microstructure that remains homogeneous, but it causes a strong degradation in the elastic properties and a drop of ≈90% in its conductivity. The incorporation of 1 M LiTFSI in PEG creates instead a weakly interconnected sphere network with poor elastic properties but still preserves ≈22% of the ionic conductivity. In contrast with the other polymers, the solution of the separated phases showed a porous monolithic microstructure that guarantees high mechanical stability and a percolation network that, backfilled with the ionic liquid, retained 30% of its original conductivity.

Relevant properties, reported in Table 3, highlight how the segregated solution outperformed the other multifunctional configurations, retaining up to 0.3% of the pure electrolyte conductivity and showing how this concept could be a suitable option for the next generation of solid multifunctional polymers.

An alternative idea to realize solid polymer electrolytes came from Ihrner et al. [84], who decided to add solvents for plasticizing PEG-based electrolyte. In this way, it was possible to create a solid polymer with its mechanical integrity, but in which, at the same time, the solvent gives the diffusive properties of a fluid. Different types of solvents and polymer compositions have been investigated via electrical impedance spectroscopy (EIS) and dynamical mechanical analysis to determine both the conductivity, the glass transition temperature, and the elastic modulus of the new material. Noteworthy results in Table 3 show a substantial improvement of the electrolyte conductivity without any detrimental effect on its mechanical properties.

Starting from these encouraging results, Ihrner et al. [85] extended the idea of applying the reaction-induced phase separation technique to manufacture biphasic electrolytes. More in detail, these materials were obtained via a UV-induced polymerization process capable of materializing a microscopically homogeneous system that works as an ionic membrane and, at the same time, has a solid connection. A further advantage of this approach lies in the opportunity of having a homogeneous low viscous fluid before curing that can be infused and cured directly with the reinforcements. This approach was then used to obtain a prototype of a 0.05 mm thin half-cell with T800HB-6000-40 carbon fiber embedded in the AB/0.65 SBE. Using a lithium metal negative electrode, the half-cell was lithiated and delithiated at a current rate of C/20 and, after an initial capacity fade due to the formation of the SEI, the half-cell showed a stable cycling behavior with a specific capacity of ≈170 mA h/g. The mechanical test showed an elastic modulus of the SBE of 730 MPa while the transverse elastic modulus of the carbon fiber half-cell was 3.1 GPa, confirming a good adhesion between the SBE and the fibers.

A direct improvement to this technique was realized by Johannisson et. al. [18], who optimized the liquid formulation of the SBE by adding a small amount of dithiol monomer (DODT). This additive guarantees a more homogeneous and slightly looser network structure, but it reduces the pot life of the compounds. Thin SBEs were realized to determine the intrinsic properties of the new electrolyte formulation, while composite laminae with unidirectional carbon fibers and a copper collector were realized as half-cell prototypes. As a term of comparison, some laminae were also produced with the same compound of the SBE but without the liquid phase. The measured conductivity of the proposed SBE reached values of 0.32 mS/cm, while its elastic modulus was of 690 MPa ≈1/4 of the monophasic structural polymer. The half-cell was then joined with a positive lithium metal electrode and a glass microfiber separator for producing a pouch cell. The latter was galvanostatically charged and discharged between 0.002 and 1.5 V vs. Li/Li^+^ for 10 cycles at a current rate of ≈C/10. The results showed a cell-specific capacity of 232 mA h/g with an initial drop of ≈100 mA h/g. Mechanical tests reported a value of the longitudinal elastic stiffness of 52 GPa, while the transverse one was of 1.7 GPa; the shear modulus was, however, of 1.5 GPa. From the comparison of these data with the data obtained with the equivalent monofunctional lamina, a reduction of ≈50% in the transverse modulus and a reduction of ≈20% in the shear modulus were observed. In the longitudinal stiffness, a difference of ≈10% only was recorded, and the SEM images of the fracture surfaces revealed that the formation of the SBE with the proposed process was not affected by the presence of the carbon fibers and that adhesion between the constituents ensured remarkable mechanical properties.

Notwithstanding the notable results obtained with the UV-initiated polymerization-induced phase separation technique, this approach has a limitation: it cannot be used for the production of non-transparent and thick SBEs. Starting from this consideration, Schneider et al. [19] demonstrated the possibility of manufacturing SBEs and unidirectional laminae electrodes via the thermally initiated polymerization-induced phase separation (PIPS) process. In this framework, the liquid electrolyte was mixed with bisphenol A dimethacrylate in a ≈40/60 wt% ratio; the compound was then placed in a mold and transferred in a preheated oven to undergo a curing cycle. Three different curing temperatures were investigated 70 °C, 80 °C, and 90 °C, and results were compared with a UV polymerized specimen. The results of the electrochemical cycling of the unidirectional (UD) half-cell confirmed that thermal cycling is suitable for the fabrication of structural batteries. A structural battery was assembled using the 80 °C thermally cured SBE that was cycled at a current rate of 0.17C. A good capacity of retention and a stable cycling behavior with a specific capacity of around 110 mA h/g were measured.

In a recent publication, Asp et al. [89] presented full-cell structural batteries with enhanced multifunctional properties. Two types of device were realized co-curing a battery-grade single-side LiFePO_4_-coated aluminum foil and a T800 fiber tow used as anode. Two types of glass fiber separators with different thicknesses were used to avoid any short circuit. The battery cell stack was then placed inside a pouch laminate bag to avoid contact with the atmosphere, and it was impregnated with a phase-separated SBE. The solid electrolyte was realized using the technique presented in Schneider et al. [19] by mixing 50:50 wt% of a liquid electrolyte solution made from the mixture of LiBoB and LiTf at concentrations of 0.4 and 0.6 M, respectively, in EC:PC 1:1 *w*/*w* (50:50 wt%) and a monomer bisphenol A ethoxylate dimethacrylate. The pouch bag was then vacuum heat sealed and thermally cured at 90 °C for one hour. Further details of the materials used for these structural batteries are described in Table 2. Tensile tests were conducted to characterize the elastic properties of the laminate in both the longitudinal and the transverse direction. A micro-tester was used to perform the activity on 30 × 3.3 mm^2^ (length × width), and the test-applied strain was calculated from the crosshead displacement in compliance with ASTM-D3379 [90]. The electrochemical tests, however, were performed by means of repeated galvanostatic charge and discharge cycles. The highest values of specific capacity were found for the batteries realized with the 50 μm glass fiber pain-weave separator that reached 23.6 W h/kg for the whole battery and 106.0 W h/kg for the active material only, at a discharge rate of 0.005 C with a nominal voltage during discharge of 2.8 V. The same cell design was also the one with the best mechanical performance, with an elastic modulus of 24.5 GPa in the 0 direction of the plain-weave composite and 13.3 GPa in the transverse direction.

The concept of phase separation introduced by Gienger et al. [83] was recently reinterpreted by Beringer et al. [91] and Lee and coworkers [26], who designed and 3D-printed optimized polymeric microstructures with enhanced multifunctional performance. Based on the know-how developed during prior studies, Beringer et al. [91] investigated the definition of an engineered structural electrolyte solution in which the involved phases were distinctly segregated to reduce at the minimum the risk to compromise their properties, and the microstructure was geometrically optimized to maximize the properties of interest in the multi-phase system. This philosophy led the group to use additive manufacturing techniques to produce epoxy-based Maxwell trusses with high structural efficiency to host liquid electrolytes (see Figure 24a). The compressive properties of the proposed configurations were investigated in compliance with ASTM-D695 [92], while the ion conductivity was measured encapsulating the unit-cells in a silicon mold and back-filling it with the liquid electrolyte. Results were compared with FEM analysis, where the shear modulus of the structure was evaluated as well. Four different interpenetrating Maxwell truss structures with different solid fractions ranging from 22% up to 84% were filled with two types of ionic liquid, the 1 M LiTFSI in PC and the 6M sodium hydroxide (KOH) in water, to evaluate their electrochemical performance. The electrochemical samples were manufactured by tightly housing the cells in a silicon test chamber and compressing against the free parallel faces of the truss two copper foils before backfilling the cell with the electrolyte. Results showed how these configurations outperform all the other precedent solutions to realize structural polymer electrolytes. Better multifunctional capabilities were recorded for the samples with 42% and 63% of the solid phase. The first one retained ≈0.2% of the mechanical properties and 40% and 64% of the LiTFSI/PC and the KOH/H2O conductivity, respectively. The other one kept ≈0.3% of the compressive modulus and ≈20% of the ionic conductivity. Notwithstanding the improved performance, this solution has not yet been capable of satisfying the multifunctional requirements of obtaining values of elastic modulus and conductivity comparable to the state-of-the-art structural polymers and conventional organic electrolytes set to 1 GPa and 1 mS/cm, respectively.

An analogous solution was also proposed by Lee et al. [26], who created a numerical tool for the evaluation of the optimal multifunctional configurations using a density-based method based on solid isotropic material with penalization model (SIMP) [93]. A multiobjective function based on a weighted sum approach of the normalized stiffness and ionic conductivity was used to evaluate the multifunctional performance of the biphasic solution. A RESNET [94] framework was applied for the numerical estimation of the electric performance, while the mechanical behavior of the unit-cell microstructures and multi-cell specimens were studied using a finite element approach considering two separate loading conditions: the uniaxial compression (see Figure 24b) and the simple shear (see Figure 24c). In contrast to the work shown by Beringer et al. [91], this activity proposed a solution where the ionic conductivity was anisotropically optimized along the direction of the ion flow, guaranteeing a greater margin in the structural stability for the same electrical performance. Optimal configurations were in this way numerically defined for each loading condition, and four different multifunctional unit-cells were manufactured. The cells with a different geometry but the same volume fraction of solid set to 0.5 were mechanically tested under compression and shear while the ionic conductivity was measured after backfilling the empty part of the cells with a 3 M KCl-based liquid electrolyte. The electrolyte was carefully selected to assure that there were no electrochemical reactions between the acrylonitrile butadiene styrene (ABS), copper electrodes and the electrolyte. Test specimens were mounted in a specially designed test rig and immersed in the KCl solution. Normalized results, reported in Figure 25, proved the capability of topological optimization as a valid support for multifunctional materials. Despite the lack of any electrodes, the morphological optimized structural electrolyte is a valid alternative to explore for the realization of future structural batteries.

The increasing interest in developing SBEs in combination with carbon fibers for multifunctional composites has also recently raised the relevance of their adhesive properties. For this reason, Xu et al. [95] first carried out an investigation of the mechanical characterization of the fiber/SBE interface. Two different carbon fiber systems were coupled with two SBEs and one monofunctional vinyl ester resin to manufacture micro-droplet samples and three-point bending UD specimens. The first tests were performed to provide a qualitative estimation of the interfacial shear strength (IFSS), while the other one gave an estimation of the composite apparent transverse tensile strength. The measured values showed in SBEs a reduction of ≈25% and 35%, respectively, for the two investigated properties in comparison with the monofunctional solutions. These results, corroborated with SEM fractography, confirmed that the liquid phase of the SBE was in contact with the carbon fibers, but, at the same time, the solid phase was sufficiently attached to guarantee good mechanical adhesion.

### 4.4. Modeling

Since the early 2000s, several experimental activities have been carried out to characterize and improve the mechanical and electrochemical properties of multifunctional composites; however, the development of theoretical and numerical tools to guide and support this progress has not been so flourishing yet. The mathematical problem, due to the several physical mechanisms involved such as electrochemistry, solid mechanics, and thermodynamics, presents a complex and coupled nature that makes the full problem difficult to solve.

Notwithstanding the fact that the first work dedicated to the topic of multifunctional composites dates back to the mid-2010s, a first pioneering theoretical study of the effect of lithium intercalation on carbon fiber particles was performed by Botte [96] in 2005. In this paper, two different approaches were used to model the lithium-ion intercalation in a carbon fiber particle: one approach neglecting the volume change induced by the lithiation and the other one including it. The numerical results, compared with experimental analyses of cyclic voltammetry tests, showed immediately the necessity to consider the coupling of the phenomena involved to obtain accurate results.

In a more recent work, in light of the experimental results obtained by Kjell et al. and Jacques and coworkers [57,58,60], Pupurs and Varna [97] implemented a numerical method to evaluate crack formation and propagation during lithium intercalation of a carbon fiber surrounded by an infinite source of ions. Thanks to this hypothesis, the lithium-ion diffusion equations were studied via a thermo-mechanical analogy, and FEM linear elastic stress analyses were used to calculate the crack propagation according to the J-integral theory. The fiber was considered infinitely long and the 2D transverse problem was studied only. Simple analyses revealed that the stress distributions obtained may induce the formation of radial cracks during deintercalation that may deflect into an arc crack during the following intercalation cycles.

A more accurate physical interpretation of this problem was then provided by Xu et al. [98,99], who investigated from a numerical point of view the electrochemical and mechanical performance of the 3D structural battery proposed by Asp and coworkers [53]. In this study, both the mass transport in the active material and in the electrolyte, the electronic conduction, and the electrochemical reactions on the surface of the active material were taken into account simultaneously and reproduced in micromechanical models while the effect of the temperature in the whole process was neglected. The main purpose was the determination of the deformation field of the battery components due to the Li-ion diffusion and the characterization of the charge rate effect on the mechanical stress distribution. Moreover, a sensitivity study on the Li-ion diffusion coefficient was performed for different charge and discharge current rates. The model represented a three cylindrical structure composed of a carbon fiber as the negative electrode, coated by an SPE and surrounded by an enriched polymeric matrix that acted as the positive electrode. The results highlighted how the fiber was subjected to radial compression both during charge and discharge, while the hoop and axial stresses were negative only during charging and switch to positive under discharging. The matrix shrinkage had instead a positive effect in reducing the possibility of damages in the coating, but it also induced tensile hoop and axial stress that can lead to matrix cracking. In the follow-up work, Xu and Varna [100] made a detailed numerical analysis of the stress field and crack propagation in unidirectional carbon fiber structural 3D batteries during the lithiation and delithiation phases (Figure 26 and Figure 27). A battery with a capacity ratio of positive- to-negative electrode *R**PN* of 0.92 and a fiber volume fraction of 0.338 was analyzed. Both the matrix and the fiber swelling and shrinkage mechanisms were taken into account, and the lithiation process was described as presented in the previous work [98,99]. The 2D models of the 3D solid battery were initially considered as stress-free in their delithiated configuration, and the evolution of the stress during the whole process was observed. As shown in Figure 26 and Figure 27, extracted from the original paper, all the stress components were the highest at full-charge condition, making this instance as the most critical part of the whole process. From the analysis of the stresses, the radial ones constantly had a compressive nature; hence, in a crack-free status, any fiber-coating debonding can nucleate. The presence of high hoop tensile stresses, however, with a peak close to the fiber coating was considered as the most harmful for the composite structural integrity for the nucleation of radial cracks. The tensile axial stress, as well, could produce a multiple fiber fragmentation process, but it was not studied in this work. Radial cracks propagation, however, was deeply analyzed via finite element analyses. The study, performed on several geometric fibers distributions representing a repeating unit in a UD composite, revealed that the cracks tended to propagate. Longer radial cracks, especially when the crack tip had almost reached the next fiber, induced an increase in the radial stresses that increased the probability of fiber debonding. This failure mechanism is particularly undesirable since it reduces battery connectivity and opens new surfaces for side reactions.

More recently, Xu and Varna [20] extended the previously presented analysis to the study of crack propagation in a [0/90] laminate. The presence of the 0 laminae with homogenized properties induced a stress field that promoted high tensile radial stress and shear stresses at the fiber coating interface. This stress field increased the probability of a debonding that tends to propagate in a mixed-mode. Moreover, contrary to what was observed for the UD in the previous work, the thermo-mechanical analyses revealed that the composite system cool down at the end of the curing process produced a substantial increase in the stress field that could not be neglected. In a parallel activity, Carlstedt et al. [101] performed a numerical study on the evolution of the elastic properties of a 3D structural battery. The effect of the SOC on the mechanical properties of the 3D battery was investigated using an analytical model of a three-phase system composed of carbon fiber coated by solid polymer electrolyte embedded in a homogenized bicontinuous polymer doped with lithium iron phosphate particles. The study readopted the concept proposed by Marklun et al. [102] for tri-phasic materials to the 3D battery, endowing it with SOC dependent parameters. The transverse properties such as *E*2 and *G*23 and the in-plane shear *G*12 were those most affected by the state of charge. The longitudinal elastic modulus *E*1, however, was almost insensitive to this factor. A sensitivity analyses of the main relevant parameters showed a high sensitivity of the composite elastic properties to the fiber volume fraction and solid polymer electrolyte stiffness.

In the following study, Carlstedt and Asp [103] proposed a semi-analytical model for the analysis of the stress field in a 3D composite battery. The model included in a coupled way the effect of shrinking and expansion produced by lithium diffusion and the heat generated during the electrochemical cycling. Due to the high electrical resistance of the constituents, only the Ohmic heat generation was considered. The thermal effect was studied for different charge and discharge rates and for different sizes of the lamina. As an example, a temperature increase up to 35 °C was detected for a current rate of 1 C that had to be included both for the induced thermal stress and for the effect of the temperature on the elastic properties of the constituents.

Although most of the recent work focused on the study of 3D models, a part of them investigated the single constituents of laminated composites. In a recent work, Tu et al. [104] developed a numerical tool for the generation of bicontinuous 3D micro-models with the aim of evaluating the effective stiffness and ion conductivity. The numerical tool was capable of generating realistic 3D microstructures with different shape and porosity (φ ∈ [0, 1], with φ = 0 for fully liquid material and φ = 1 for continuous solids), and it could be used as a virtual material testing device to lead the generation of optimized bicontinuous electrolytes. Three-dimensional micro-models were generated, and the homogenized elastic properties together with the ionic conductivity were measured independently. Results, compared with the experimental data, confirmed that microstructure with a high and low level of porosity tended to have good performance in one effective property only. Materials with an intermediate porosity (φ ≈0.5), generally defined as trabecular, and imperfect trabecular structures, instead, highlighted good bi-functional properties.

Dionisi et al. [105], however, presented an analytical model to study the deformation and stresses in laminated structural batteries. The method, based on the classical laminated plate theory, was endowed with a 3D stress shape function for the study of unbalanced and unsymmetrical laminates. Anodes and cathodes were considered as transversally isotropic carbon fiber laminates endowed with specific longitudinal and transverse linear expansion coefficients dependent on lithium concentration; the separator, however, was considered as a plain-weave glass fiber lamina. All constituents were assumed as embedded in a solid polymer electrolyte that allowed the movement of the ions. The material shrinkage and swelling induced by lithium intercalation were studied by using a thermal expansion analogy and no extra thermal effect was included. Different battery lay-ups were analyzed to determine both the global deformation and the interlaminar stress of the laminate induced by the lithium-ion intercalation in order to evaluate the risk of delaminations. The results were validated via finite element simulations and revealed interesting outcomes for the design of the structural batteries. The most relevant was the fact that a symmetric structural laminate with a stacking sequence [Anode/Separator/Cathode]_S_ had much lower interlaminar stress than an opposite configuration with the cathode on the outer shell side.

More recently, inspired by the vast amount of work on traditional lithium-ion batteries [106], a thermodynamically consistent framework for investigating electro-chemo-mechanical-coupled models of laminated structural batteries was presented by Carlstedt et al. [107]. The model was applied in the study of a negative half-cell with a lithium metal counter electrode, as experimentally investigated by other authors [18,19,85], while the separator was excluded for simplicity. The problem was solved using the commercial FE software COMSOL Multiphysics in a 2D domain with a 3D stress field taking into account different boundary conditions and electrochemical couplings. Two types of electrochemical cycling were studied: the galvanostatic, with a constant discharge current, and the potensiostatic, with a constant potential, while from the mechanical point of view a generalized plane stress condition was enforced. The overall behavior was in line with the experimental data. Other conditions here investigated involved the study of the plane strain and the generalized plane strain under galvanostatic control. The same mechanical boundary condition was then used to investigate two different types of coupling: one-way coupling, where the electrochemical problem is solved independently and the results are used as input for the mechanical one, and two-way coupling with a full interconnection. Results obtained from a single galvanostatic discharge revealed a change in the stress state proportional to the lithiation of the anodes. The study was concluded with an investigation of the out-of-plane mechanical boundary conditions showing a high correlation between the electrochemical performance and the mechanical conditions in line with the experimental evidence reported by Jacques et al. [108], highlighting the importance of the electro-chemo-mechanical interactions for this type of problem.

## 5. Strategies towards the Future

The strategy for the future is based on overcoming the most hindering problems toward structural batteries’ application. The following difficulties have to be addressed: (1) safety due to the use of lithium and liquid/gel flammable electrolytes; (2) capacity loss due to lithiation/delithiation leading to poor cyclability; (3) mechanical properties degradation with cycling; (4) low specific energy; (5) low curing temperatures (<130 °C); (6) poor ionic conductivity; (7) limited range of working temperatures.

The all-solid-state solution seems to be the best to address the safety problem. Nonetheless, all-solid-state batteries are also plagued by specific bottlenecks: (1) slow kinetics of ion diffusion in solid-state electrolytes and the transport of ions across the solid- solid interfaces; (2) chemical instabilities at Li metal-solid electrolyte and high voltage cathode-solid electrolyte interfaces; (3) local mechanical and structural instabilities in solid-state electrolytes that fail to resist lithium dendrites and compromise safety; (4) the necessity of renewing the existing Li-ion assembly lines and equipment, which is an additional impediment for fast commercialization of all currently available all-solid-state solutions.

Recently, we have been developing structural batteries. Our strategy is based on the use of a ferroelectric-electrolyte non-flammable composite (A2.99Ba0.005ClO, A = Li, Na) that polarizes spontaneously below 170 °C [109,110]. This spontaneous polarization is due to the ferroelectric character of the electrolyte and adds to the electrostatic storage in the battery cell. The electrolyte is able to plate alkali metal (Li or Na) on the electrodes, and therefore to discharge featuring a thermodynamic equilibrium plateau corresponding to the reduction of Li^+^ to Li on a relay with low chemical potential such as sulfur [111], carbon, or copper. This electrochemical reaction adds to the capacity of the cell.

The goal is to not use alkali metals while assembling the structural batteries but in- expensive conductors, such as zinc and carbon as electrodes/current collectors. We highlight that a protective layer of electrolyte naturally covers the plated metals.

The battery cells can be cured at temperatures as high as 250 °C, depending on the polymers used in the composite cells and performs between −20 °C and 180 °C; it is eco-friendly, inexpensive, and safe. We expect that any mechanical properties shortcomings will be overcome by a thorough investigation of suitable polymers, fibers, and surface and cell engineering.

## 6. Conclusions

Structural batteries have been in the last decade one of the most appealing technological solutions to reduce the weight, the volume, and the consumption of modern electric vehicles and devices. This relevance has pushed many researchers to investigate the physical chemistry, the opportunities, the production techniques, and the margins of optimization in this emerging field.

This paper presented a detailed review of the recent advances in structural power composites. A recent evolution in both integration techniques for off-the-shelf lithium-ion batteries in composite structures and multifunctional materials for power applications has led to many works in this field.

The experimental activities in the field demonstrate and suggest remarkable improvements in structural battery components in recent years. It is particularly important to highlight the electrochemical characterization of the carbon fibers as structural anodes, the adoption of doping techniques to realize carbon fiber-based cathodes, and the tuning of epoxy systems to realize stiff and ion-conductive structural solid electrolytes. However, the full- and half-cell batteries reported in the literature show remarkable differences in mechanical and/or storage performance. Because this field involves highly complex physical phenomena, the development of reliable analytical and numerical frameworks are expected to aid in the understanding and optimization of the interaction between all battery constituents, and possibly help to overcome the current problems that have been hindering structural batteries application, such as safety problems, capacity and mechanical properties degradation with cycling, low specific energy and energy density, poor manufacturing conditions, poor ionic conductivity, and limited range of working temperatures. In addition, the use of a novel ferroelectric-electrolyte non-flammable composite is seen as a promising solution to some of these problems.

## Figures and Tables

**Figure 1 molecules-26-02203-f001:**
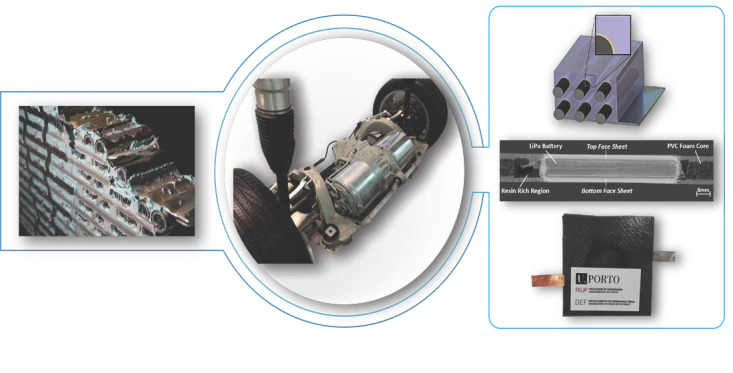
Structural power composites as an alternative to battery pack dead weight.

**Figure 2 molecules-26-02203-f002:**
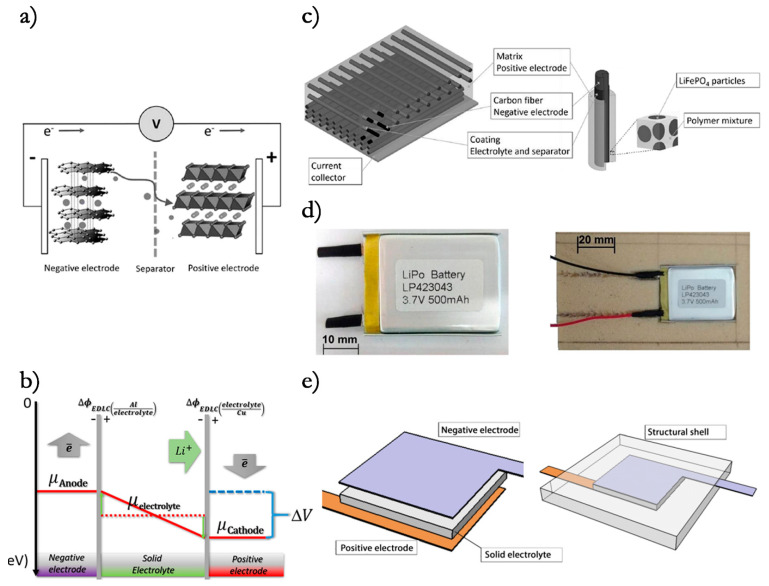
Structural power composite principles: (**a**) Lithium-ion battery [20] and (**b**) Solid state battery [21]. Structural power composite applications: (**c**) Multifunctional material with structural battery electrolyte [20], (**d**) Multifunctional systems with market available lithium-ion batteries embedding [10] and(**e**) All-solid-state structural battery.

**Figure 3 molecules-26-02203-f003:**
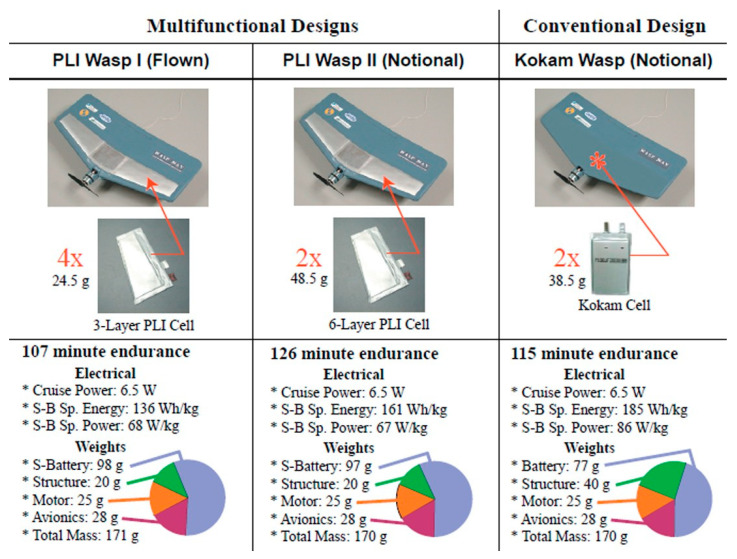
Unmanned aerial vehicles (UAVs) improved performance via multifunctional structure [23].

**Figure 4 molecules-26-02203-f004:**
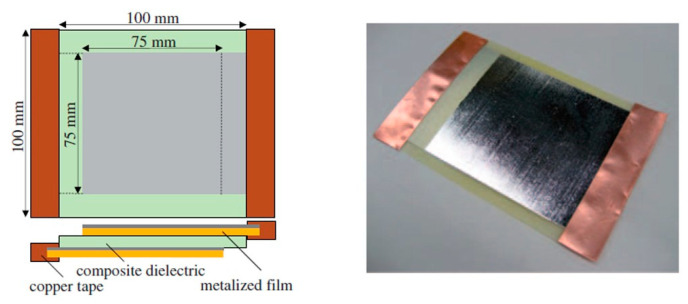
Structural capacitors [32].

**Figure 5 molecules-26-02203-f005:**
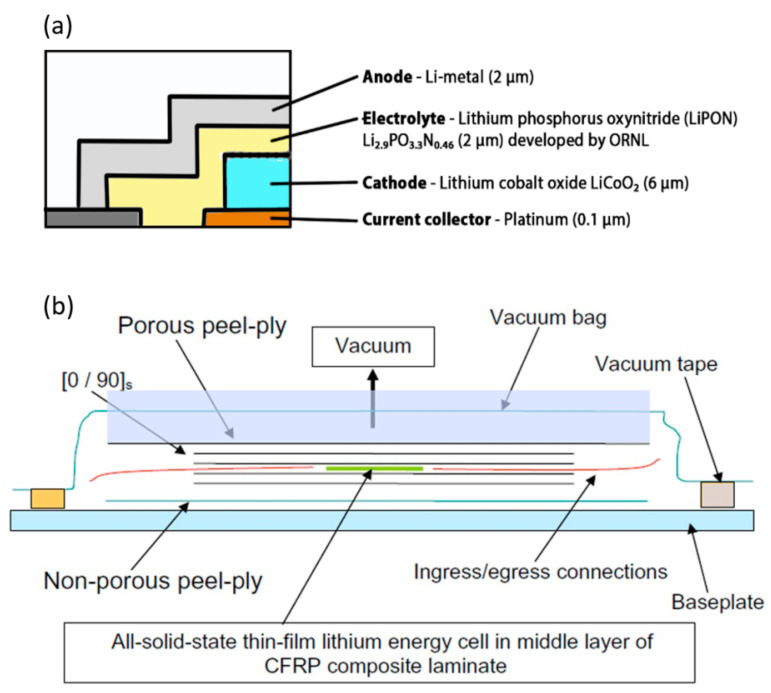
Integration of thin-film Li-ion energy cells in carbon fiber reinforced laminates. (**a**) All-solid-state thin-film lithium cell. (**b**) Schematic layup of thin-film lithium energy cell embedding in CFRP [6].

**Figure 6 molecules-26-02203-f006:**
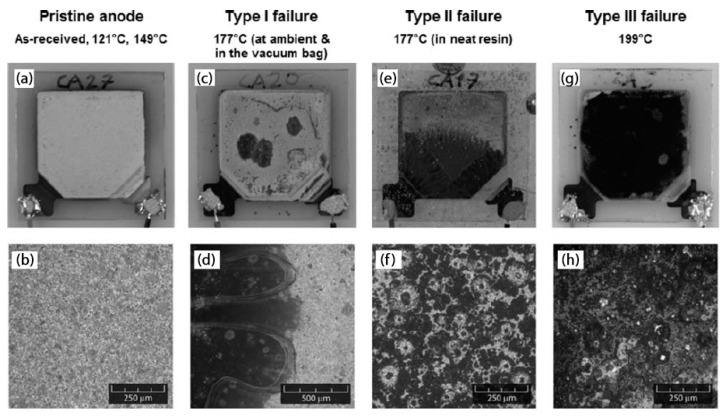
Battery thermally tested and surface magnification showing: (**a**–**b**) no degradation, (**c**–**d**) gray spots, (**e**–**f**) sealing/anode reactions and (**g**–**h**) melted anode [25].

**Figure 7 molecules-26-02203-f007:**
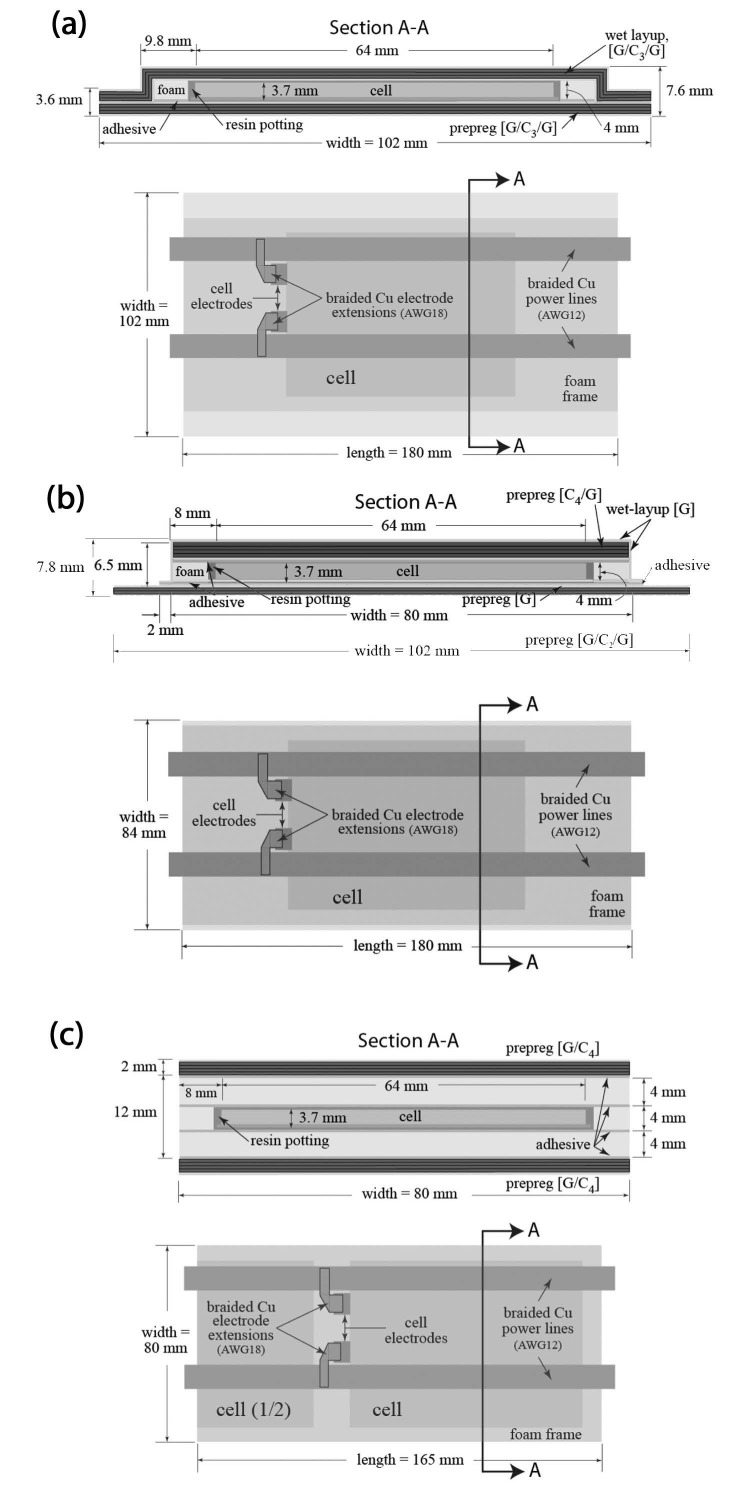
Structural battery sandwich configuration for marine systems [5]. (**a**) Integrated SB laminate. (**b**) Modular SB stiffener. (**c**) SB sandwich.

**Figure 8 molecules-26-02203-f008:**
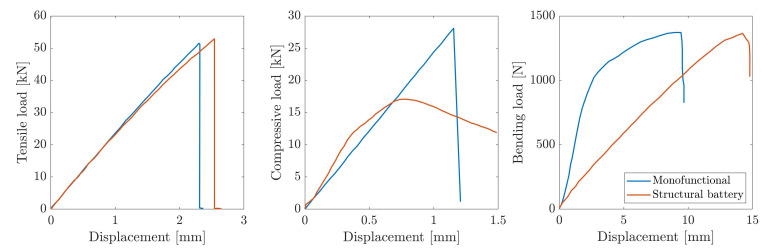
Effect of cell embeddings in sandwich panel mechanical performance [38].

**Figure 9 molecules-26-02203-f009:**
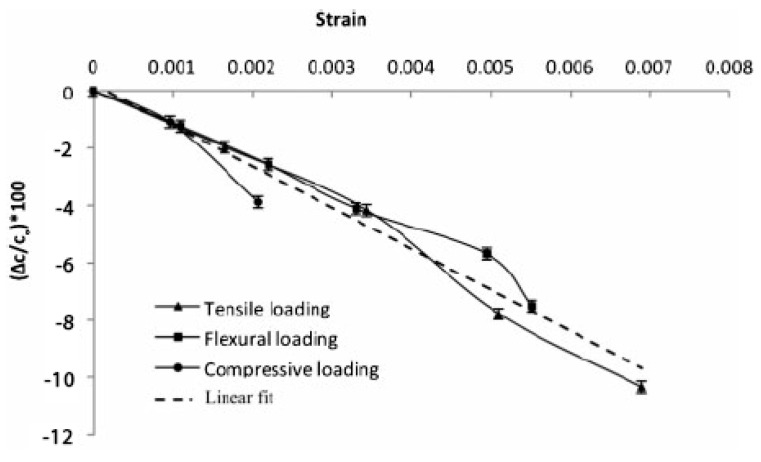
Strain-induced battery capacity loss [38].

**Figure 10 molecules-26-02203-f010:**
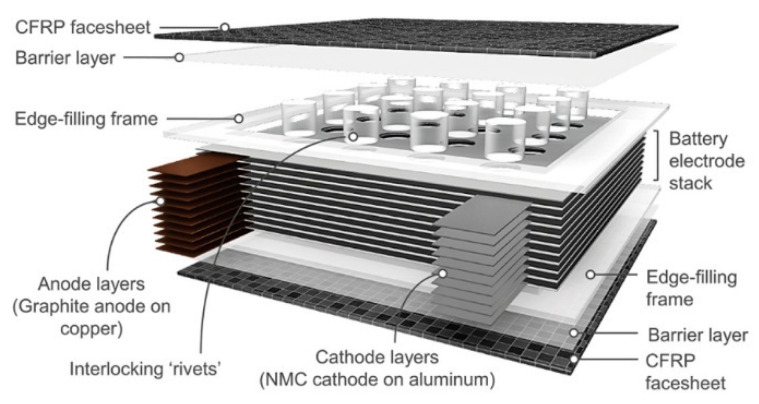
Multifunctional energy storage composite with interlocking rivets [27].

**Figure 11 molecules-26-02203-f011:**
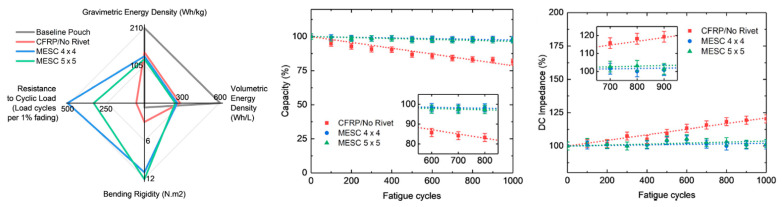
Interlocking rivet energy storage composites performance [27].

**Figure 12 molecules-26-02203-f012:**
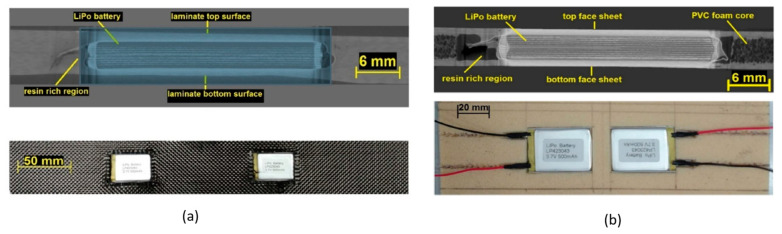
Structural composite battery configurations. (**a**) Laminated battery [8]. (**b**) Sandwich battery [10].

**Figure 13 molecules-26-02203-f013:**
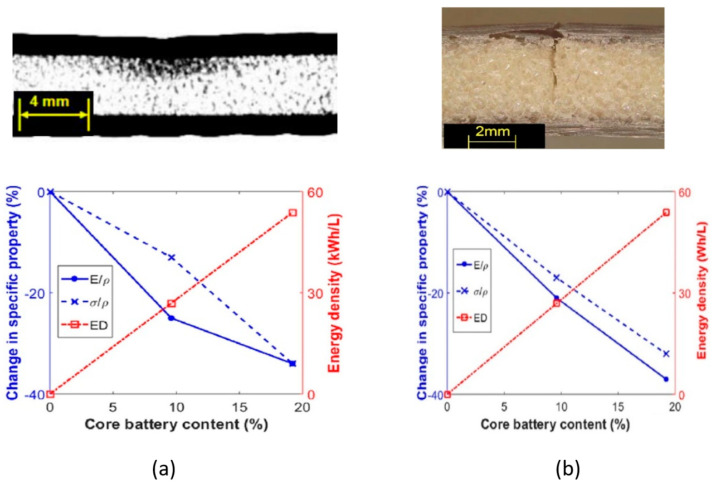
Failure modes of sandwich structures under three-point bending [10]. (**a**) 90 mm span. (**b**) 200 mm span.

**Figure 14 molecules-26-02203-f014:**
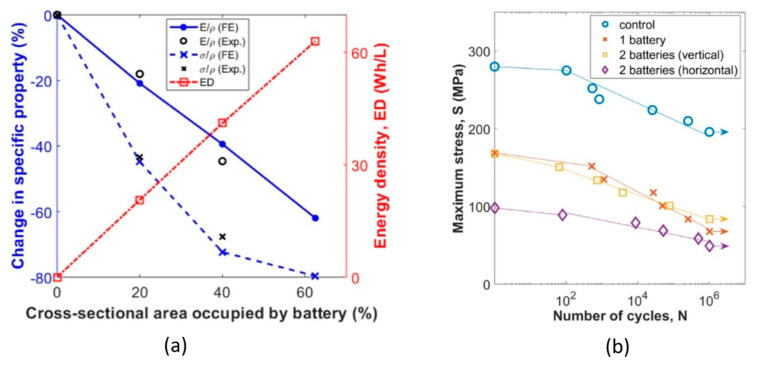
Compressive behavior of laminated batteries [9]. (**a**) Quasi-static performance. (**b**) Fatigue.

**Figure 15 molecules-26-02203-f015:**
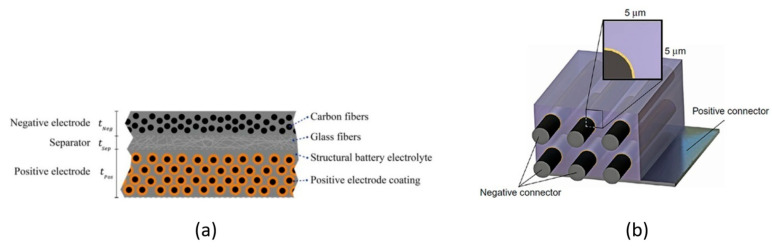
Conceptualization of composite structural batteries. (**a**) Laminated structural battery from Johannisson et al. [28]. (**b**) 3D structural battery from Carlson et al. [24].

**Figure 16 molecules-26-02203-f016:**
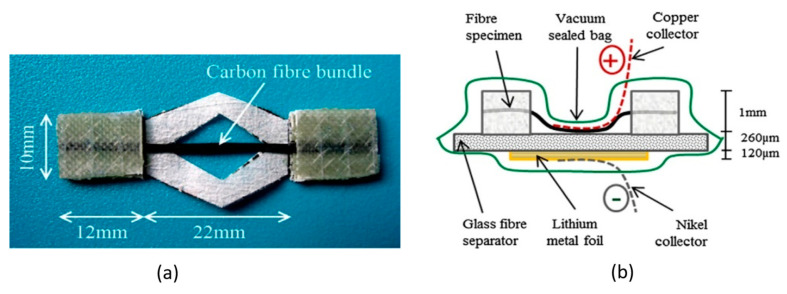
Fiber set-up [58]. (**a**) Tensile specimen. (**b**) Electrochemical cell.

**Figure 17 molecules-26-02203-f017:**
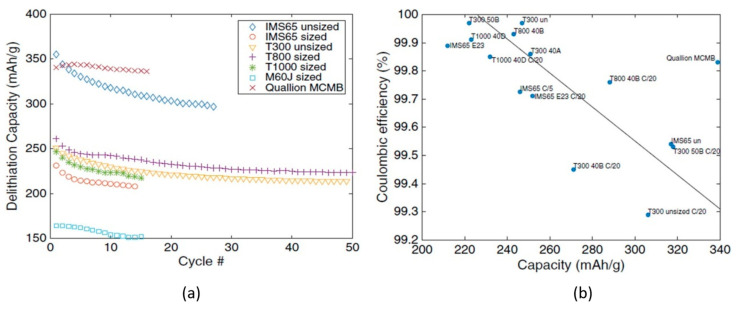
Polyacrylonitrile (PAN)-based fiber electrical performance and comparison with MCMB [61]. (**a**) Delithiation capacity vs. cycle number for PAN-based fiber cycled at C/10. (**b**) 10th cycle Coulombic efficiency against specific capacity.

**Figure 18 molecules-26-02203-f018:**
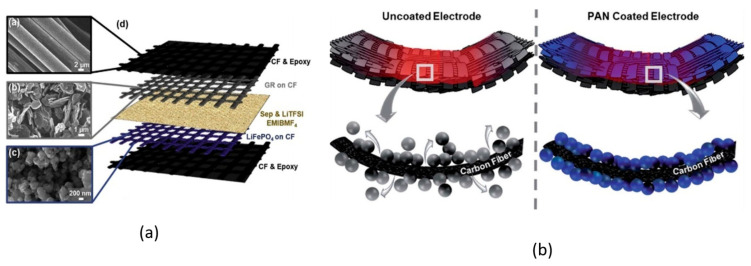
Structural battery with engineered interfaces from Moyer et al. [29,63]. (**a**) Carbon fiber battery layout. (**b**) PAN coating process.

**Figure 19 molecules-26-02203-f019:**
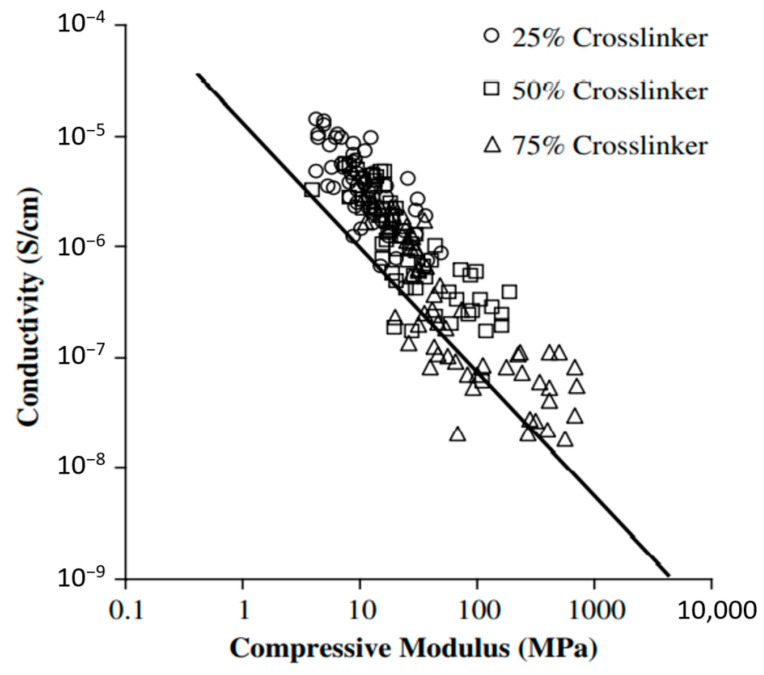
Conductivity vs. compressive modulus for structural polymer electrolytes investigated by Snyder et al. [76].

**Figure 20 molecules-26-02203-f020:**
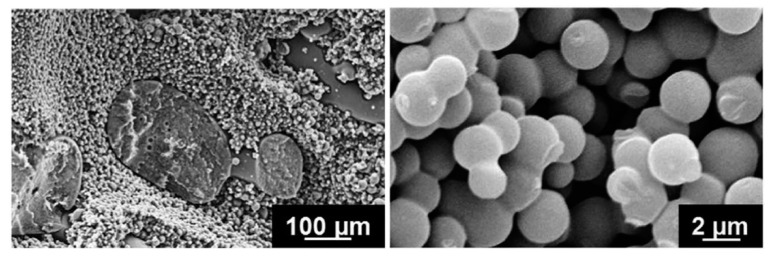
MTM57 50% SEM micrographs with a porous epoxy phase and connected spherical nodules from Shirshova et al. [79].

**Figure 21 molecules-26-02203-f021:**
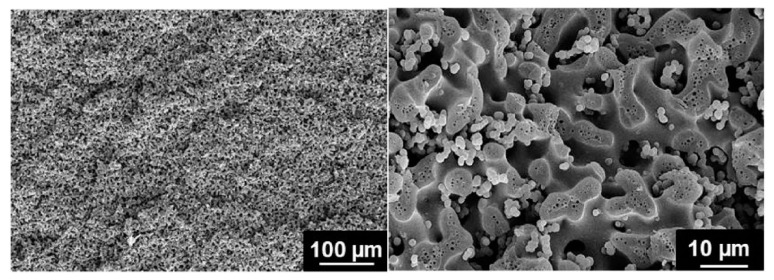
MTM57 50% with 4.6 mol/L of LiTFSI SEM micrographs with a more homogeneous biphasic microstructure from Shirshova et al. [80].

**Figure 22 molecules-26-02203-f022:**
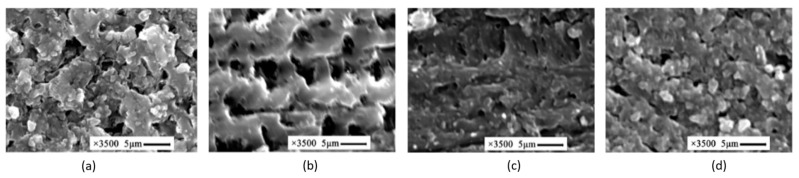
Organically modified silicates (OLS) sensitivity of epoxy-based electrolytes microstructure from Yu et al. [81]. (**a**) 0% OLS. (**b**) 2.5% OLS. (**c**) 5.0% OLS. (**d**) 7.5% OLS.

**Figure 23 molecules-26-02203-f023:**
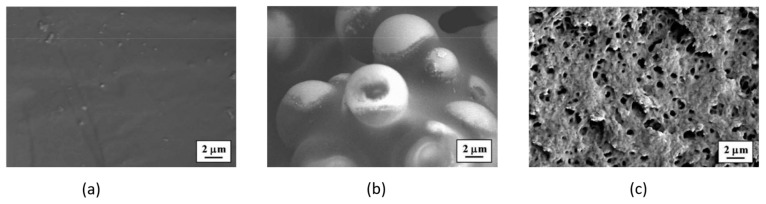
Microstructure for EPON 828/PACM-based multifunctional polymers at 65% of electrolyte content from Gienger et al. [83]. (**a**) 1 M LiTFSI in PC. (**b**) 1 M LiTFSI in PEG. (**c**) Segregated 1M LiTFSI in PC.

**Figure 24 molecules-26-02203-f024:**
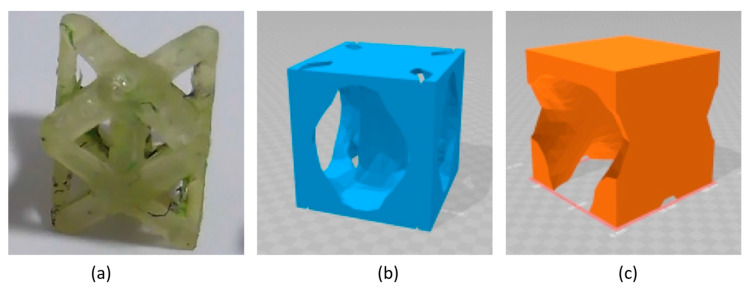
Multifunctional unit-cell prototypes proposed by Beringer et al. and Lee et al. [26,91]. (**a**) Maxwell truss [91]. (**b**) Compressive unit-cell [26]. (**c**) Shear unit-cell [26].

**Figure 25 molecules-26-02203-f025:**
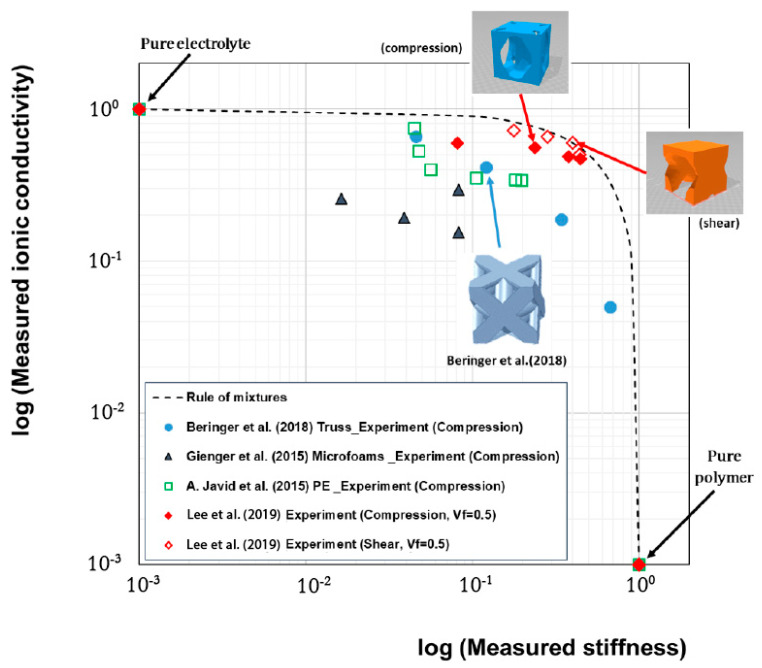
Performance of multifunctional engineered unit-cells from Lee et al. [26]. (N.B. the stiffnesses and ionic conductivities were normalized against the stiffness of a pure solid polymer and the ionic conductivity of a pure electrolyte, respectively).

**Figure 26 molecules-26-02203-f026:**
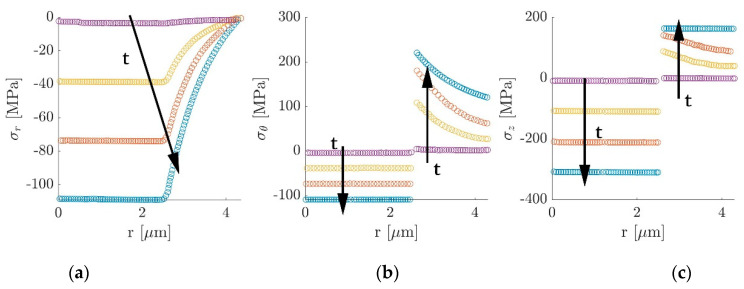
Stress distribution during charging: (**a**) radial stress; (**b**) hoop stress; (**c**) axial stress from Xu et al. [100].

**Figure 27 molecules-26-02203-f027:**
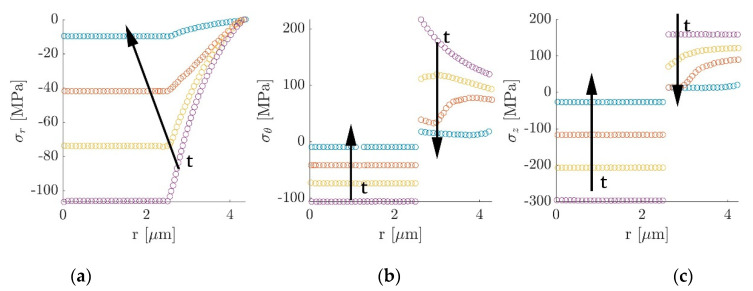
Stress distribution during discharging: (**a**) radial stress; (**b**) hoop stress; (**c**) axial stress from Xu et al. [100].

**Table 1 molecules-26-02203-t001:** Multifunctional configurations [10,23,24,25,26,27,28,29].

Multifunctional Systems		Multifunctional Materials	
[23]	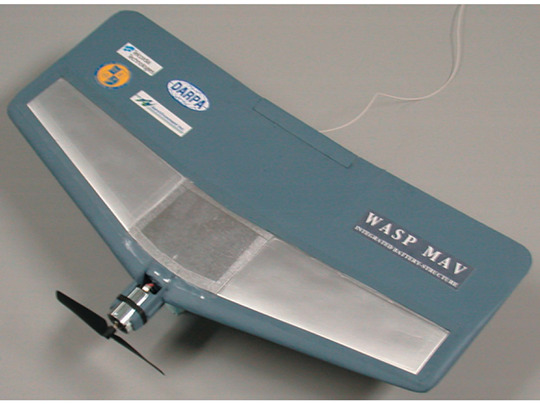	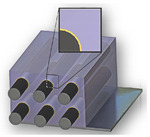	[24]
[25]	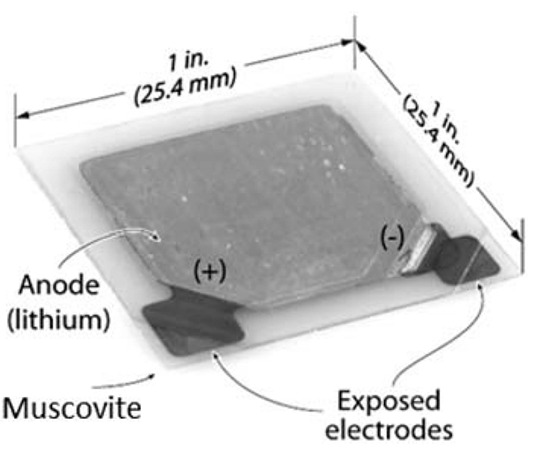	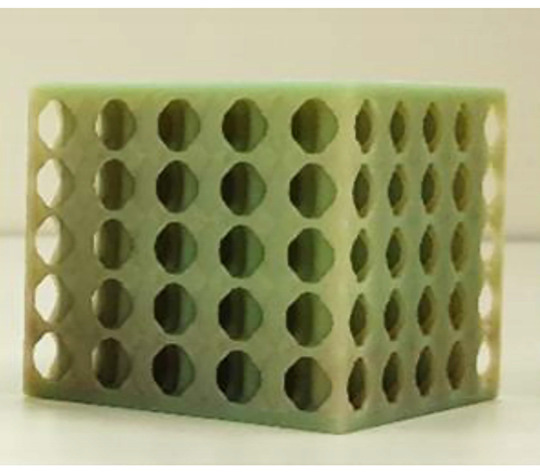	[26]
[27]	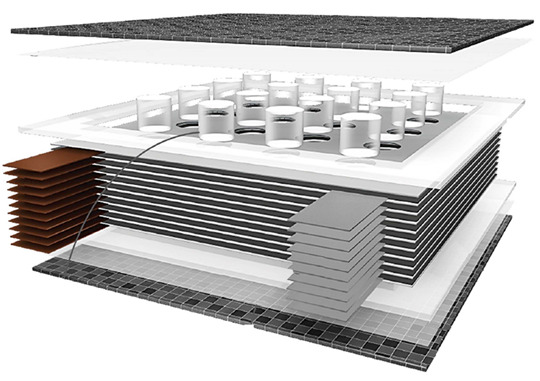	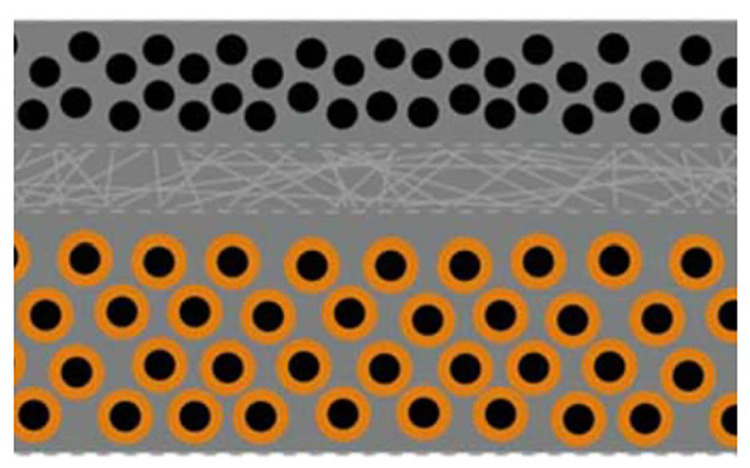	[28]
[10]	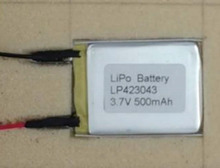	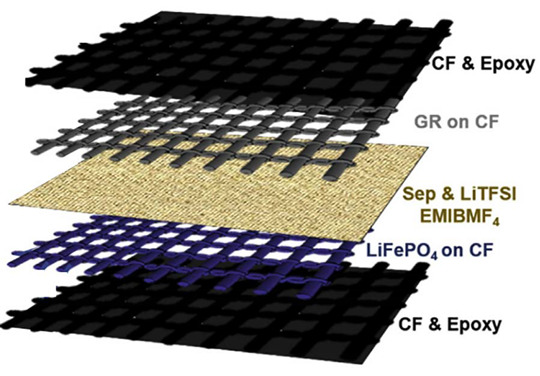	[29]

**Table 2 molecules-26-02203-t002:** Material list of structural batteries.

Ref.	Anode	Cathode	Separator	Electrolyte
**Half-cell for anode characterization**				
[58,59,60]	Lithium metal 120 µm thick	T800/IMS65	Whatman GF 260 µm thick	1.0 M LiPF_6_ in EC and diethyl carbonate (1:1 wt%)
[61]	Lithium metal	T300/T800 M60J/IMS65 T1000G	Whatman GF 250 µm thick Porosity 90%	1.0 M LiPF_6_ in EC and diethyl carbonate (1:1 wt%) 2% vinylene carbonate
**Half-cell for cathode characterization**				
[62]	Lithium metal	LiFePO_4_ powder conductive carbon black PVDF binder in different ratios on AS4 carbon fiber	Whatman GF 250 µm thick Porosity 90%	1.0 M LiPF_6_ in EC and diethyl carbonate (1:1 wt%)
**Full-Cell**				
[29]	Graphite powder < 20 µm conductive carbon black PVDF binder (80:10:10) on PAN based carbon fibers	LiFePO_4_ powder conductive carbon black on carbon nanotubes PVDF binder (65:20:5:10) on PAN based carbon fiber	Whatman GF	1.0 M LiTFSI in EMIMBF_4_ ionic liquid
[63]	PAN coated, graphite powder < 20 µm conductive carbon black PVDF binder (80:10:10) on PAN based carbon fibers	PAN coated, LiFePO_4_ powder conductive carbon black on carbon nanotubes PVDF binder (65:20:5:10) on PAN based carbon fiber	Celgard 2525	1.0 M LiPF6 in ethylene carbonate and diethyl carbonate (1:1 wt%)

**Table 3 molecules-26-02203-t003:** Structural polymer electrolytes properties.

Ref.[79]	Sample	Process	*T* [°C]	σ [mS/cm]	*T**g* [°C]	*E* [MPa]
[79]	MTM57/45% ^a^	Th120	20	0.23	111	180
[79]	MTM57/50% ^a^	Th120	20	0.13	111	150
[79]	MVR444/30% ^a^	Th100	20	0.8	-	180
[79]	MVR444/40% ^a^	Th100	20	0.07	-	490
[80]	2.3M MTM57/50% ^b^	Th100	20	0.43	111	220
[80]	3.5M MTM57/50% ^b^	Th100	20	0.12	102	180
[80]	4.6M MTM57/50% ^b^	Th100	20	0.01	104	420
[81]	E51-AG80 0.0OLS	Th150	20	0.56	47	373
[81]	E51-AG80 2.5OLS	Th150	20	0.47	41	136
[81]	E51-AG80 5.0OLS	Th150	20	0.89	37	211
[81]	E51-AG80 7.5OLS	Th150	20	0.84	33	198
[82]	E51-AG80 0.5IL	Th160	20	0.05	-	400
[82]	E51-AG80 1.0IL	Th160	20	0.1	-	200
[83]	EP-CM 1Li-PC 65% ^c^	Th80	20	0.41	23	17
[83]	EP-CM 1Li-PEG 65% ^c^	Th80	20	0.09	-	5
[83]	EP-CM 1LiPC * 65% ^c^	Th160	20	1.5	74	120
[84]	50:50 ^d^	UV-Solvent	20	0.03	−15	20
[84]	90:10 ^d^	UV-Solvent	20	0.005	29	240
[85]	A/60 ^e^	UV-PIPS	25	0.15	-	750
[85]	A/65 ^e^	UV-PIPS	25	0.21	-	530
[85]	AB/60 ^f^	UV-PIPS	25	0.11	-	730
[85]	AB/65 ^f^	UV-PIPS	25	0.19	-	550
[85]	B/60 ^g^	UV-PIPS	25	0.12	71	380
[85]	B/65 ^g^	UV-PIPS	25	0.20	72	690
[18]	≈A/65 ^h^	UV-PIPS	25	0.32	-	380
[19]	≈A/60	Th70-PIPS	25	0.19	-	440
[19]	≈A/60	Th80-PIPS	25	0.20	-	540
[19]	≈A/60	Th90-PIPS	25	0.19	-	540
[19]	≈A/60	UV-PIPS	25	0.20	-	540

^a^ The % indicates the estimated resin volume content [vol%]. ^b^ The M indicates the mol/L of LiTFSI used in the compound. ^c^ The EP-CM indicate the resin system used in this study based on EPON 828 and Amicure 4,4-Diaminodicyclohexylmethane (PACM). The 1Li-PC indicates the electrolyte tested containing 1M LiTFSI in propylene carbonate (PC), while the 1Li-PEG indicates one with 1M LiTFSI in polyethylene glycol (PEG). The * superscript indicates the multifunctional polymer with segregated phases, while the % stands for the electrolyte content in volume. ^d^ The x:y weight indicates the percentages of the monomers SR209 and SR550 used for the polymer, while the solvent composition is in both cases a solution of 0.2 g of ethylene carbonate (EC) and 0.2 g of dimethyl methyl phosphonate (DMMP). ^e^ 1 g of bisphenol A dimethacrylate with 0.6/0.65 g of lithium tri-fluoromethanesulfonate (LiTFS). ^f^ 0.5 g of bisphenol A dimethacrylate and 0.5 g of bisphenol B dimethacrylate with 0.6/0.65 g of lithium tri- fluoromethanesulfonate (LiTFS). ^g^ 1 g of bisphenol B dimethacrylate with 0.6/0.65 g of lithium tri-fluoromethanesulfonate (LiTFS). ^h^ No liquid phase added.

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
