# Peer review of "Structural Batteries: A Review"

_molecules, 2021, doi:10.3390/molecules26082203_

Round 1

Reviewer 1 Report

The manuscript titled "Structural batteries: a review" can be accepted for publication after the following minor revisions.

  1. Fix typos and grammatical mistakes present in several places of the manuscript.
  2. Emphasize the importance of Li-ion batteries compared to other energy storage devices before discussing the structural Li-ion batteries
  3. Extend the discussion on structural batteries to post Li-ion batteries such as Na-ion batteries.

Author Response

Replies added in the attached file.

Reviewer 2 Report

Very interesting and well written review. As minor comment, I suggest to add a paragraph/figure with a clear comparison of the different presented systems.

Author Response

Replies added in the attached file.

Reviewer 3 Report

This review presents an overview of the recent development of the battery assembly and provides a cue for a possible alternative configuration. The Manuscript is presented clearly and contains detailed information. Therefore I suggest publishing after a minor revision. The comments are

  1. I suggest making a Table showing the list of materials used to assemble battery.
  2. Similarly, the Authors are suggested adding the table showing list of solid electrolytes uses so far with conductivity
  3. Energy density or specific energy? Make consistent.
  4. The influence factors affecting battery life should be discussed from the perspectives of design, production and application.
  5. The growth of dendrites that limit the battery life should be discussed in the review.
  6. The authors discussed the PAN-based Carbon fibers as the anode materials (https://www.sciencedirect.com/science/article/pii/S002197971930743X; https://www.sciencedirect.com/science/article/pii/S2352152X2031906X). Carbon fibers are in general brittle and have low flexibility and mechanical properties. How these properties affect during assembly.

Author Response

Replies added in the attached file.
